# Identification of rainfall thresholds for debris-flow occurrence through field monitoring data.

Elena Ioriatti[1], Mauro Reguzzoni[2], Edoardo Reguzzoni[2], Andreas Schimmel[3], Luca Beretta[4], Massimo Ceriani[5], Matteo Berti[1]

[1]Department of Biological, Geological, and Environmental Sciences (BiGeA), University of Bologna, Bologna (BO), 40126, Italy
[2]Hortus S.r.l., Gallarate (VA), 21013, Italy
[3]Andreas Schimmel – Alpine Monitoring Systems (ALMOSYS), Mkt. Piesting, 2753, Austria
[4]Direzione Generale Territorio e Sistemi Verdi, Lombardia Region, Milano (MI), 20124, Italy
[5]Geologist, Cardano al Campo (VA), 21010, Italy

*Correspondence to:* Elena Ioriatti (ele.ioriatti@gmail.com)

**Abstract.** Defining rainfall thresholds for debris-flow initiation typically requires numerous past events, but many catchments lack sufficient historical records. This study introduces a method based on monitoring data, effective even with few observed debris flows. The approach relies on rainfall measurements and images from a simple, low-cost monitoring station. The method was developed in the Blè catchment, in the central Italian Alps. An algorithm based on a minimum inter-event time was used to automatically identify rainfall events. Throughout the monitoring period, which included both wet and dry conditions, stream's hydrological regime was classified into four categories according to water level and sediment transport. Each rainfall event was linked to the catchment response, and an intensity–duration scatterplot was generated with events categorized accordingly. A threshold was defined using Linear Discriminant Analysis, treating events that triggered regime changes as positive and others as negative. This threshold offers insight into catchment behaviour and can be rigidly shifted upward to isolate only debris-flow-triggering events. Results show good discriminative ability and reliable performance in distinguishing regime-changing events. Finally, the study explores how the threshold is affected by the rain gauge's location and by the method used to define rainfall events.

## 1. Introduction

Debris flows are among the most hazardous types of landslides due to their high velocity, destructive potential, and sudden onset (Iverson, 1997; Jakob & Hungr, 2005). These fast-moving mixtures of water and sediment can travel long distances along steep channels, often causing severe damage to infrastructure and posing significant risks to human life (Guzzetti et al., 2005; Hilker et al., 2009).

One of the most effective methods for mitigating debris-flow risk is the definition of rainfall thresholds (Martelloni et al., 2012; Pan et al., 2018; Papa et al., 2013). In such systems, precipitation data recorded by rain gauges are continuously compared against predefined critical thresholds to predict the potential initiation of debris flows. Various approaches have been proposed in the literature to establish such thresholds, including empirical methods (e.g., Aleotti, 2004; Cannon et al., 2008; Ceriani et al., 1994; Coe et al., 2008; Deganutti et al., 2000; Gariano et al., 2015; Hirschberg et al., 2021) and, less frequently, physically based modelling approaches (e.g., Berti and Simoni, 2005; Crosta and Frattini, 2003; Hoch et al., 2021; Iverson, 2000; Montgomery and Dietrich, 1994; Wei et al. 2024). Although the latter are grounded in a physical representation of processes, their practical application is often challenging and uncertain due to their reliance on detailed knowledge of physical and hydrological parameters, which are often only approximately known (Gariano et al., 2020; Melchiorre & Frattini, 2012). Empirical methods, on the other hand, rely on the analysis of rainfall events that have

historically triggered debris flows but require a substantial number of such events to define reliable thresholds. This dependence on extensive historical data poses a major limitation, particularly when attempting to establish thresholds at the catchment scale, where such data are often lacking.

The reliability of rainfall thresholds defined with an empirical approach can be influenced by several sources of uncertainty, including the spatial distribution of rain gauges, the criteria used to define individual rainfall events, and the temporal resolution of rainfall data. Marra et al. (2016) and Nikolopoulos et al. (2014) highlighted that rain gauge networks with limited spatial coverage can underestimate rainfall during convective storms. This may lead to thresholds that do not accurately reflect triggering conditions. Another key source of uncertainty lies in the method used to identify discrete rainfall events from continuous data (Melillo et al., 2015). A common approach is to use a minimum inter-event time, but there are still no clear criteria for determining its optimal duration (Dunkerley, 2008). The temporal resolution of rainfall data is another important factor, as coarse resolution has been shown to systematically underestimate depth–duration thresholds (Marra, 2019; Gariano et al., 2020).

In this study, we propose an alternative approach to define intensity–duration rainfall thresholds, which is based on the use of monitoring data collected over a relatively short period and does not require a large number of debris-flow events. The method relies on data acquired through relatively low-cost sensors and a lightweight, easy-to-install monitoring station. This station was located on the stream bank within an Alpine catchment. The monitoring data provided a good understanding of the catchment's hydrological response, allowing the identification of a lower threshold associated with increases in stream water level and sediment transport that may serve for pre-alert purposes.

To overcome the limitation posed by the small number of debris-flow events, we use two complementary strategies. First, we consider not only triggering but also non-triggering rainfall events, applying statistical analysis to distinguish between the two classes. While this practice is established at regional scales, it has rarely been implemented within catchment-scale monitoring. Second, we draw on the larger set of high-flow and sediment-transport events to establish a robust lower threshold, which then serves as a reference for isolating debris-flow conditions and defining the debris-flow threshold.

In addition, the study explores the uncertainty in threshold definition associated with two key factors: the spatial location of the rain gauge and the duration of the inter-event time used to separate rainfall events.

## 2. Study area

The Blè catchment is located in Val Camonica, Lombardia Region, in the Central Italian Alps. It covers a drainage area of about 2 km², with elevations ranging from 2,540 m to 770 m a.s.l. (Fig. 1). The associated debris-flow fan extends from the basin outlet to its confluence with the Oglio River at 360 m a.s.l. and has an areal extent comparable to that of the contributing basin. The large size of the fan reflects the significant debris-flow activity that has characterized the basin over thousands of years, likely combined with the accumulation of a massive post-glacial landslide that reached the opposite valley side. The 3.7 km long Blè Torrent flows along the right margin of the fan in its upper part, before crossing the fan's centre further downstream and joining the Oglio River. Within the basin, a complex deep-seated landslide is located on the left side of the hydrological network, while on the right side lies another landslide, known as "La Tavola", whose toe intersects the Blè stream (Fig. 1). It is highly likely that both current landslides originated from a single, massive post-glacial translational landslide evolved into a rock avalanche (estimated at over 300 million m³), which was subsequently incised by the erosion of the Blè stream. At present, La Tavola landslide exhibits slow movement and supplies material to the channel. However, increased debris-flow activity could further destabilize the slope by eroding its toe, potentially triggering a collapse and resulting in channel damming. According to the Geological Map of Italy

1:50,000 (Sheet 057 "Malonno"; ISPRA, 2012), the lithology of the basin is predominantly composed of Triassic carbonate rocks, including limestones and breccias, as well as platform deposits in the uppermost part of the catchment. The middle and lower portions of the basin are partially covered by scree debris (Fig. 2).

In the Blè catchment, debris flows are primarily triggered by surface runoff at the contact between bedrock and talus. Runoff generated on steep, rocky slopes mobilizes loose debris accumulated in hollows or at the base of cliffs, initiating downslope transport through progressive entrainment. Figure 1 illustrates the headwater sub-catchment that concentrates surface runoff and drains towards the debris-flow initiation zone, located near its outlet. In some cases, however, debris flows initiate along the channel downstream due to the mobilization of material resulting from undercutting at the toe of La Tavola landslide.

Two major debris flows were documented in the 20th century, occurring in August 1950 and September 1960. Five events have been recorded in the past seven years alone, indicating a probable increase in event frequency and suggesting a potential shift in the stability conditions of the basin and the associated stream system (Table 1). This recent intensification of debris-flow activity has drawn the attention of regional authorities and emphasized the necessity of implementing a monitoring and early warning system.

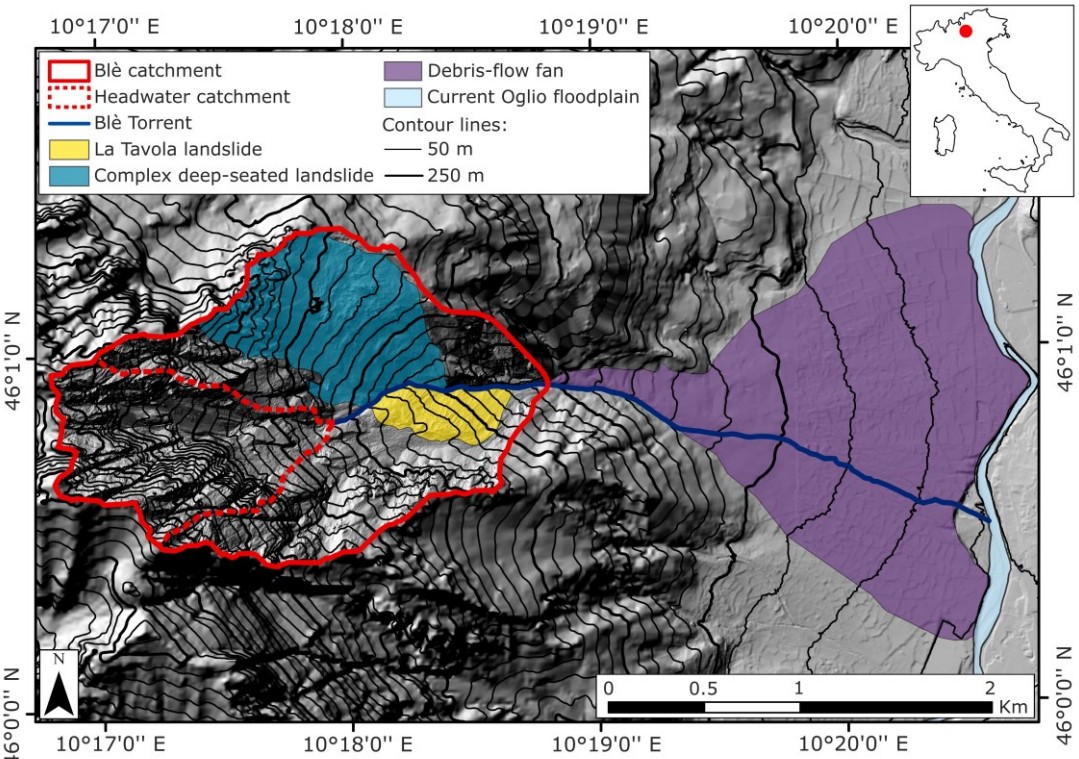

**Figure 1: Map of the main geomorphological features of the Blè catchment. The base map is adapted from the Geoportal of Lombardia Region.**

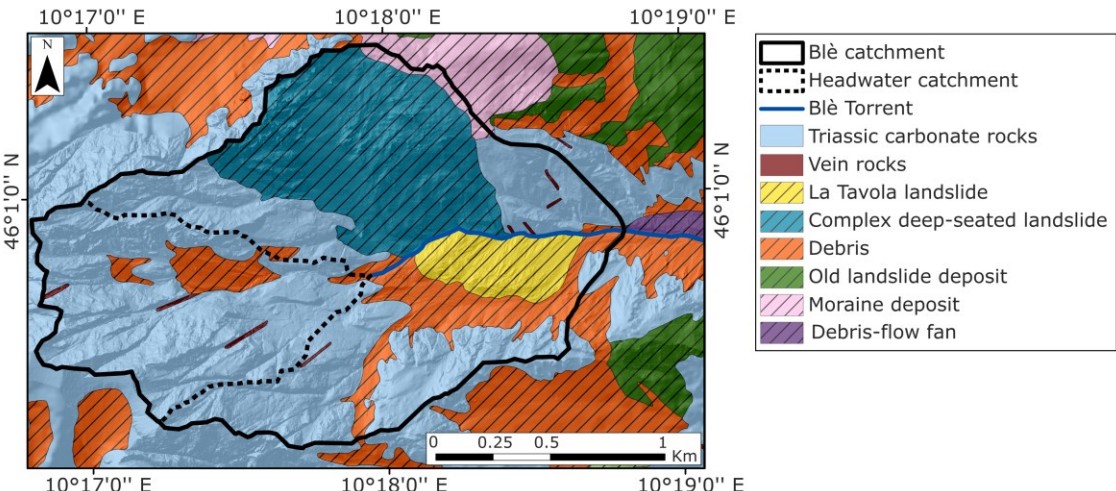

**Figure 2: Geological map of the Blè catchment, showing lithology and deposits. The map is adapted from the Geoportal of Lombardia Region.**

**Table 1: Dates of debris-flow events and their estimated volumes.**

| Date | Volume (m³) |
|---|---|
| 25 Aug 2018 | 20,000 |
| 27 Jul 2019 | 25,000 |
| 6 Aug 2019 | 100,000 |
| 16 Aug 2021 | 50,000 |
| 22 Oct 2022 | ≈ 50,000 |

## 3. Methods

### 3.1 Monitoring system

The monitoring station was installed on 14 July 2021 on the left bank of the Blè Torrent, at an elevation of 735 m a.s.l. (Fig. 3). This station, hereafter referred to as "UNIBO", was designed to be lightweight, rapidly deployable in the field, and cost-effective. To meet these design criteria, it does not include heavy support structures and is therefore dismantled in autumn to prevent potential damage from winter conditions.

UNIBO station is equipped with a tipping-bucket rain gauge with a sensitivity of 0.2 mm, and a time-lapse camera programmed to capture images of the channel at 15 min intervals (Fig. 4). It also features a MAMODIS debris-flow detection system (Schimmel et al., 2018) and a DataCube seismograph (DiGOS, 2020), equipped with three vertical geophones with a natural frequency of 4.5 Hz. However, data from these two instruments are not considered in the present study.

The monitoring equipment was destroyed by a debris flow on 16 August 2021, due to the failure of the channel bank. However, a portion was successfully recovered, allowing for the retrieval of data recorded during the event. The system was subsequently reinstalled on 23 September 2021 in the same channel reach but relocated a few metres further from the bank.

The following year, in spring 2022, additional six monitoring stations were installed by Hortus S.r.l., as part of an integrated monitoring and early warning system funded by Lombardia Region (Fig. 3). Unlike the UNIBO station, these additional stations were professionally engineered and required substantial financial support. Four stations (H1, H2, H3,

H4) became operational between 11 June and 1 July 2022, while the last two (H5, H6) became operational on 23 July 2023. Rain-gauge data from the four uppermost Hortus stations (H1 to H4; Fig. 5) were used in this study to analyse spatial uncertainty in threshold definition.

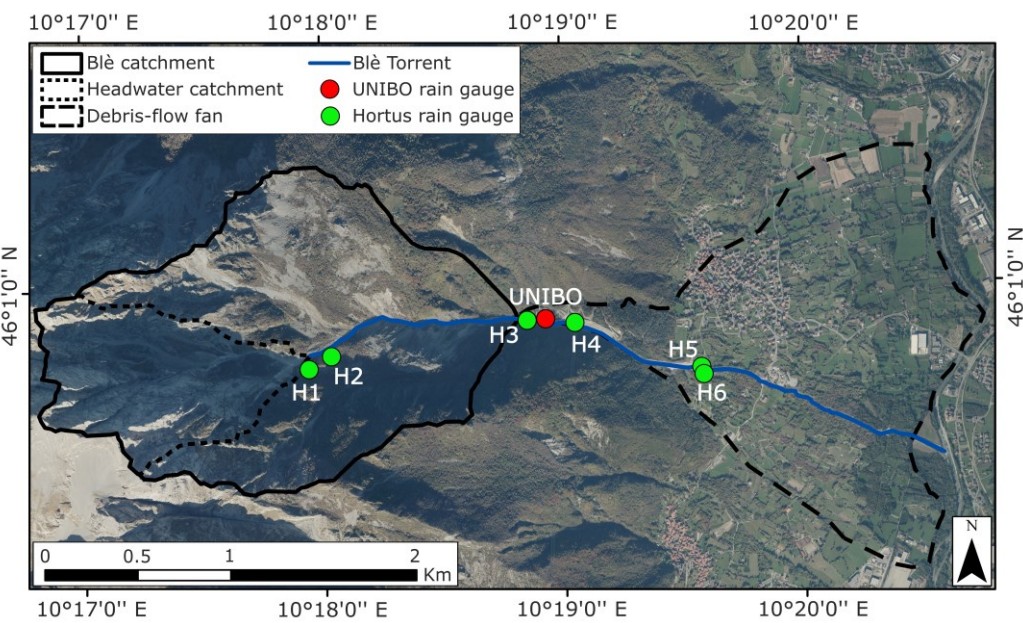

125

**Figure 3: The monitoring and warning system installed in the Blè catchment (Base map: Orthophoto AGEA 2021–WMS, Lombardia Region).**

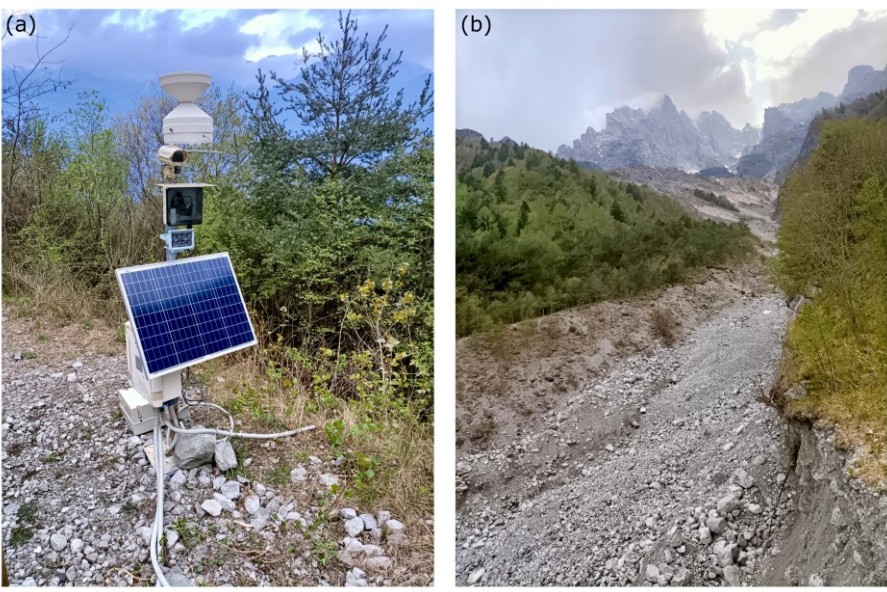

**Figure 4: (a) Photograph of the UNIBO monitoring station. (b) View of the channel from the position of the monitoring station in April 2023.**

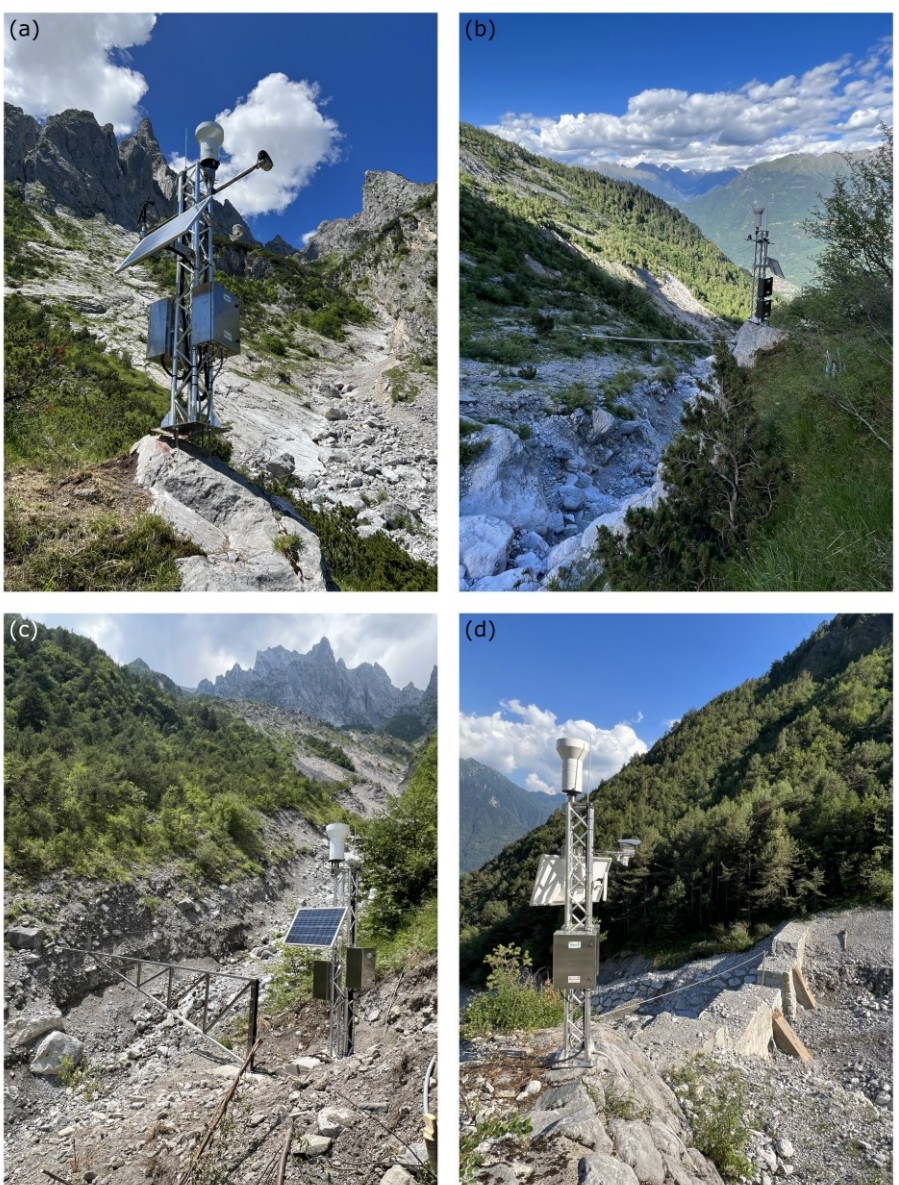

**Figure 5: Photos of the four uppermost stations of Hortus S.r.l. network for monitoring and warning a) H1, b) H2, c) H3, d) H4.**

## 3.2 Identification of rainfall events

The identification of rainfall episodes plays a fundamental role in defining rainfall thresholds, as the precipitation parameters used to establish these thresholds directly depend on how events are delineated (Abancó et al., 2016). In some cases, identifying the triggering rainfall is straightforward, for example, when an intense precipitation event is preceded by a prolonged dry period. In others, rainfall sequences consist of multiple bursts, making it challenging to determine when one event ends and the next begins (Berti et al., 2012).

Common methods described in the literature for defining the start and end of a rainfall event involve setting thresholds for the duration and precipitation amount of the dry period separating consecutive events (Postance et al., 2018). These thresholds help identify the no-rainfall interval, ensuring a clear distinction between successive precipitation episodes. Typically, short inter-event periods are applied to shallow landslides and debris flows, which respond quickly to rainfall

(Abancó et al., 2016; Staley et al., 2015; Deganutti et al., 2000; Badoux et al., 2009; Coe et al., 2008), whereas longer periods are used for deep-seated landslides occurring in fine-grained materials (Berti et al., 2012; Brunetti et al., 2010).

In this study, rainfall events were identified using a minimum dry interval of two consecutive hours without recorded precipitation. This approach is comparable to the criterion adopted in the Dolomitic area by Berti et al. (2020), where a one-hour no-rainfall interval was used to separate events. This choice was motivated by similarities between the Blè catchment and Dolomitic basins. These include limestone cliffs with coarse, unconsolidated debris at their base, comparable initiation mechanisms through channel runoff, and rapid hydrological responses to rainfall.

Rainfall data with a temporal resolution of 5 minutes were collected between late spring and early autumn in 2021, 2022, and 2023. The specific dates for the analysed periods are provided in Table 2. To investigate the sensitivity of the threshold to the criteria used for event identification, the same procedure was repeated using no-rainfall periods of varying durations: specifically, 1, 3, 4, 5, 6, 7, 8, 12, 18, and 24 hours.

**Table 2: Summary of the periods analysed for precipitation data (UNIBO, H1, H2, H3, H4) and images (Camera) during the years 2021, 2022, and 2023.**

| Rain gauge / Camera | From | To |
|---|---|---|
| **UNIBO** | 14 Jul 2021 | 16 Aug 2021 |
| | 23 Sep 2021 | 24 Nov 2021 |
| | 9 Mar 2022 | 11 Nov 2022 |
| | 17 Apr 2023 | 25 Oct 2023 |
| **H1** | 11 Jun 2022 | 16 Nov 2022 |
| | 29 Jun 2023 | 26 Oct 2023 |
| **H2** | 11 Jun 2022 | 16 Nov 2022 |
| | 29 Jun 2023 | 26 Oct 2023 |
| **H3** | 1 Jul 2022 | 16 Nov 2022 |
| | 17 Apr 2023 | 26 Oct 2023 |
| **H4** | 23 Jun 2022 | 16 Nov 2022 |
| | 17 Apr 2023 | 26 Oct 2023 |
| **Camera** | 27 Jul 2021 | 16 Aug 2021 |
| | 23 Sep 2021 | 24 Nov 2021 |
| | 16 May 2022 | 11 Nov 2022 |
| | 19 Apr 2023 | 25 Oct 2023 |

**3.3 Classification of images and linking rainfall events to the observed channel response**

The hydrological response of the catchment to rainfall events was analysed using images captured every 15 minutes by the camera (Fig. 6). Each rainfall event was classified into one of the following classes:

C0 - Night: Images captured at night when it is not possible to observe the activity in the channel.

C1 - Low flow: The channel flow is at its baseline condition, with no noticeable increase in discharge.

C2 - High flow: The channel discharge increases compared to the low-flow condition, and the flow becomes turbid.

C3 - High flow with sediment transport: The channel discharge increases further, accompanied by visible sediment transport along the stream. Slight channel migration, caused by erosion and deposition, is also evident.

C4 - Debris flow: A rapid surge characterized by the chaotic transport of irregular clasts with sharp edges. Visible clasts typically reach few tens of centimetres in their longest dimension, although larger blocks are observed, not only at the flow front but also dispersed throughout the central body. Woody debris is occasionally present but not abundant.

ND - No data: No observations available because the camera was not working or visibility was obscured, such as during intense storms.

Event classification was operator-based and supported by a script that displays the images and automatically generates a structured dataset with time intervals and corresponding class labels. In an initial attempt, classification was carried out manually by reviewing each image, recording start and end times, and assigning class labels. This method proved inefficient and error-prone due to manual transcription. The script improved the process by providing an interface with simple controls that allowed the operator to classify each image, automatically creating a continuous time-series dataset

that recorded the class and the corresponding start and end times of each observed process. Although classification still required expert input, the script greatly reduced transcription errors and accelerated the overall workflow.

Some classification uncertainties arose during rainfall, when visibility was compromised by dense fog or water droplets on the camera lens. In such instances, it was sometimes helpful to examine images of the channel before and after the event to determine whether sediment transport, along with associated erosion or deposition, had occurred. Weather

conditions, such as sunny or cloudy days, and the time of the day could also lead to misinterpretations. For example, in shaded areas water flow was often less visible, while in bright sunlight reflections on the water surface could create the illusion of higher discharge. To address these challenges, two of the authors jointly reviewed a large number of cases and established shared criteria to distinguish actual changes in discharge from lighting or visibility artefacts. These criteria were then applied consistently across all images to ensure comparability and limit operator bias.

Data from the water level sensor were examined but proved unreliable, as the dataset contained negative values and showed inconsistencies with the observed flow dynamics. The sensor was mounted high above the channel to avoid damage during debris flows, and therefore its wide measurement cone is suitable for capturing extreme events but insufficiently sensitive to reliably detect smaller flow increases.

In a few cases, a C2 or C3 level was observed just before nightfall or early in the morning, as soon as the channel became

visible in the images. In such cases, there was uncertainty about whether the event might have reached a higher level during the night. These events were flagged in the dataset and subsequently excluded to assess their impact on intensity–duration scatterplots. The goal was to determine whether removing these uncertain events would improve the clustering of each class and make the boundaries between them more distinct. However, no significant improvement in class separation was observed, so these events were ultimately included in the dataset.

Capturing one photo every 15 minutes does not allow for precise timing of the channel response relative to the start of rainfall, but it is sufficient to describe the overall event. Each precipitation event was characterised by the highest observed class within its duration.

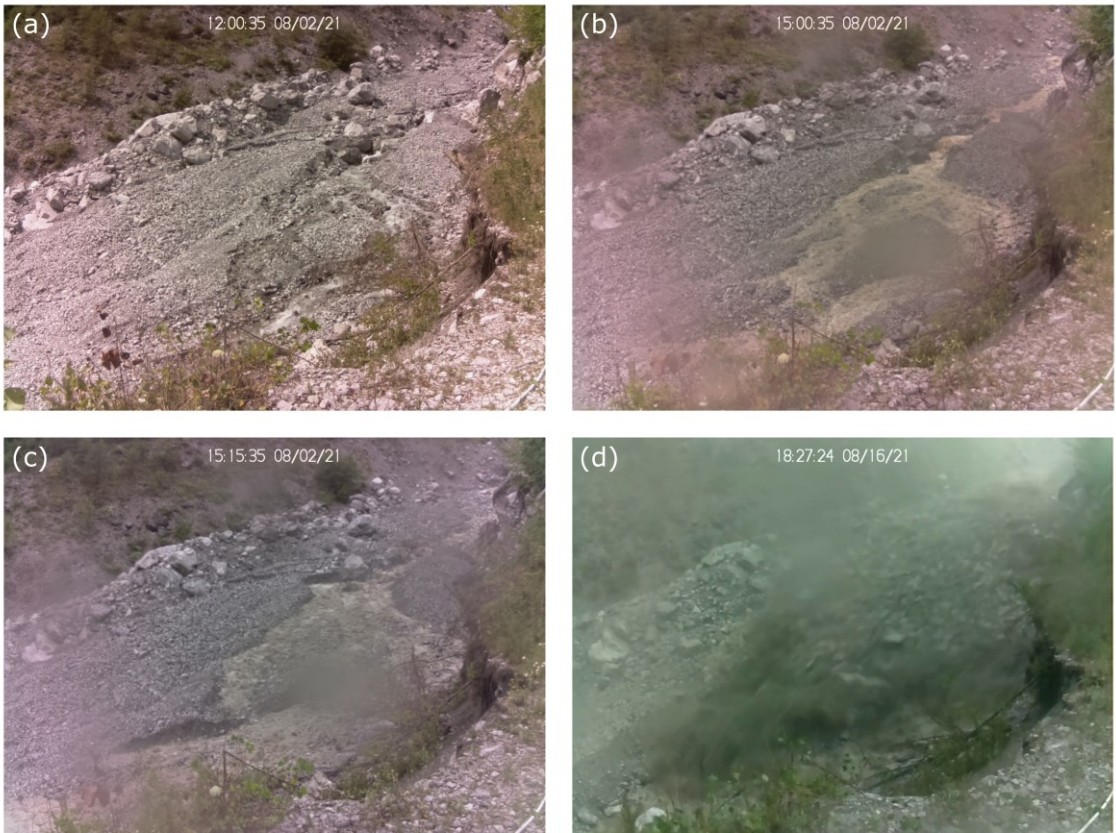

**Figure 6: Examples of channel processes captured by the UNIBO monitoring system: (a) C1 (low flow), (b) C2 (high flow), (c) C3 (high flow with sediment transport), (d) C4 (debris flow). Time is shown in UTC+2.**

### 3.4 Rainfall threshold definition using the Linear Discriminant Analysis (LDA)

Two thresholds were defined to separate the observed hydrological responses: TH1 is a lower threshold distinguishing

low flow (C1) from high flow, high flow with sediment transport and debris flow (C2, C3, C4), while TH2 is an upper threshold distinguishing debris flow (C4) from all other classes (C1, C2, C3).

To determine TH1, the method of Linear Discriminant Analysis (LDA) was applied to the dataset, treating low-flow events (C1) as non-triggering ("False") and all other responses as triggering (C2, C3, C4, "True"). LDA is a statistical method for dimensionality reduction and feature selection that identifies a linear combination of input variables to

optimally separate triggering and non-triggering classes (Fisher, 1936; Ramos-Cañón et al., 2016). In this study, we did not apply dimensionality reduction from a larger feature set; instead, rainfall duration and average intensity were defined a priori as the predictor variables for the analysis. LDA was then applied to the rainfall events, with the aim of identifying a discriminant axis that maximizes between-class variance and minimizes within-class variance, as described by the objective function J(w) in Eq. (1) (Bishop, 2006):

$$J(w) = \frac{w^T S_b w}{w^T S_w w},$$                                                    (1)

Where:

$S_b$ is the between-class scatter matrix, representing the variance between the class means.

$S_w$ is the within-class scatter matrix, representing the variance within each class.

$w$ is the linear combination vector (the discriminant direction).

For the debris-flow threshold (TH2), events classified as debris flows (C4) were considered "triggering" ("True"), while all other classes (C1, C2, C3) were treated as "non-triggering" ("False"). In this case, the LDA method was not applied due to the limited number of debris flows, which made it challenging to accurately estimate the within-class variance and class means required for a reliable discriminant axis. Additionally, the significant imbalance between classes distorts the separation boundary, as the dominance of the majority class shifts the boundary toward the minority class, reducing the model's ability to distinguish between groups. TH2 was defined by keeping the same scaling exponent β (slope) as TH1 and selecting the coefficient α that maximises the Area Under the Receiver Operating Characteristic Curve (AUC). Although assuming parallelism between TH1 and TH2 is methodologically convenient, it can be questioned from a hydrological perspective, as the rainfall intensity–duration relationship may differ between flow-depth increases and debris-flow mobilization. However, for runoff-generated debris flows, studies have shown that the two thresholds display similar slopes, at least for the short-duration events that typically trigger debris flows (Berti and Simoni, 2005; Simoni et al., 2020; Berti et al., 2020). This similarity arises because runoff generation and the mobilization of channel debris are both expressions of the same hydraulic process: the concentration of overland flow within the catchment and its transformation into channelized flow.

**3.5 Evaluation of rainfall variability across multiple monitoring stations.**

The influence of rain-gauge location on threshold definitions was assessed by comparing rainfall records from the UNIBO gauge with those from the four Hortus sites (Fig. 3). To determine whether a precipitation event recorded by different stations should be considered the same event, the following criteria were applied:

- If a rainfall event recorded at a Hortus gauge overlaps, even momentarily, with one at the UNIBO gauge, they are considered the same event.
- If multiple overlapping events from Hortus stations correspond to a single UNIBO event, the one with the highest cumulative rainfall was selected.

For each matched event, duration, cumulative precipitation, and mean intensity from the Hortus stations were directly compared to the corresponding values from UNIBO. Additionally, for each overlapping event, the minimum, maximum, and mean duration ($D_n$) and intensity ($I_n$) across all stations were calculated (Fig. 7).

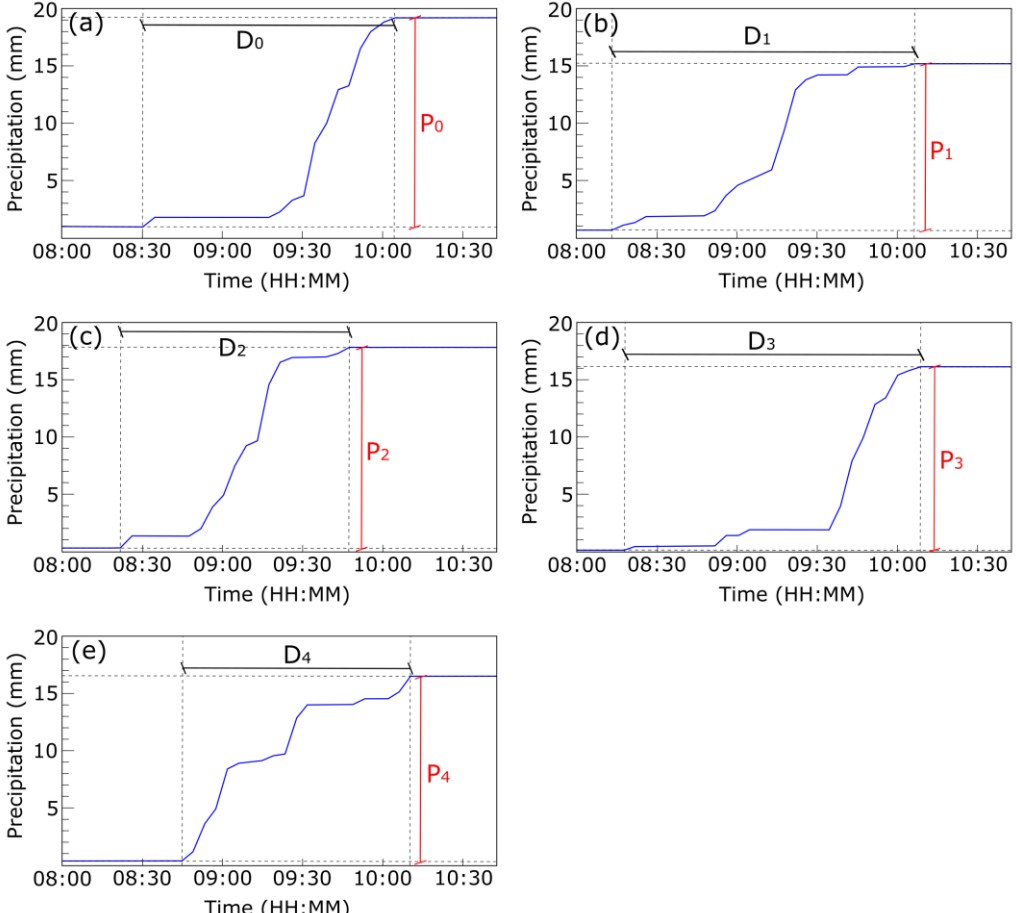

**Figure 7**: *Illustrative example showing how minimum, maximum, and mean values of duration and intensity are calculated using data collected from all rain gauges in the catchment. Subplots represent the same rainfall event as recorded at different stations: (a) UNIBO, (b) H1, (c) H2, (d) H3, and (e) H4. The following statistics were computed: min {D0, D1, D2, D3, D4}, max {D0, D1, D2, D3, D4}, mean {D0, D1, D2, D3, D4}, min {I0, I1, I2, I3, I4}, max {I0, I1, I2, I3, I4}, mean {I0, I1, I2, I3, I4}. In is calculated as Pn•Dn⁻¹.*

## 4. Results

### 4.1 Classified rainfall events

Rainfall data collection began at the UNIBO site on 14 July 2021, while measurements from the Hortus network became available in 2022, specifically from 11 June for H1 and H2, 23 June for H4, and 1 July for H3. The UNIBO rain gauge recorded two debris flows (C4), which occurred on 16 August 2021 and 22 October 2022. Additionally, it registered 9 events classified as high flow with sediment transport (C3), 21 high-flow events (C2), and 184 rainfall events that did not alter the flow regime (low flow, C1). Each Hortus gauge detected 1 debris flow, along with 4 to 6 C3 events, 12 to 14 C2 events, and 82 to 136 C1 events. Among the four Hortus stations, the lower-elevation rain gauges, H3 and H4, recorded the highest number of rainfall events. Across all locations, the proportion of overnight occurrences (C0) ranged from 30 % to 40 %. A total of 21 events at UNIBO and one at H1 could not be classified due to the absence of camera images at the time (ND; Fig. 8).

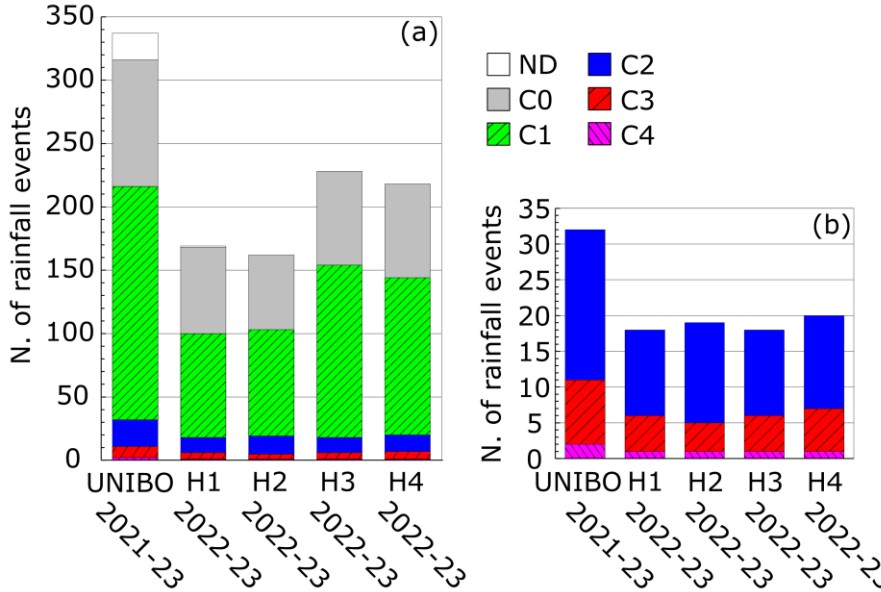

 **Figure 8: (a) Number of rainfall events recorded at each monitoring station, classified according to the observed channel response. The inset (b) zooms in on the classes with lower frequencies.**

All the rainfall events recorded by the UNIBO station are shown in Fig. 9. The different symbols correspond to the various response types described in Sect. 3.3: C0 (night), C1 (low flow), C2 (high flow), C3 (high flow with sediment transport), C4 (debris flow), and ND (no data). Distinct hydrological behaviours are expected to be well separated in the feature space; however, a certain degree of overlap is observed.

Rainfall associated with C2 and C3 response levels generally exhibited longer durations and higher intensities compared to C1 conditions, although the boundaries between classes remain poorly defined. Precipitation linked to C2 responses consistently exceeded 1.2 mm h$^{-1}$, while that associated with C3 responses was always greater than 2.0 mm h$^{-1}$. The two rainfall episodes that triggered debris flows occurred in different periods of the year and were characterized by distinct durations and mean intensities. The July 2021 event lasted 55 minutes with a mean intensity of 56 mm h$^{-1}$, whereas the October 2022 episode spanned 17 hours and 45 minutes, with a mean intensity of 5 mm h$^{-1}$ (Fig. 9).

The separation of C2, C3, and C4 events from C1 events is clearer at short durations than at longer durations. One could argue that this occurs because long-duration events include short high-intensity phases that control the hydrological response, and that it is therefore not appropriate to consider durations exceeding the time of concentration (Tc). For durations close to Tc the entire catchment contributes to runoff; consequently, at the intensity Ic associated with Tc, durations greater than Tc should not further increase discharge. One would therefore expect the threshold to be horizontal for D > Tc, indicating an approximately constant basin response.

In practice, however, rainfall is not uniformly distributed across the catchment. As duration increases, the precipitation cell is more likely to shift and cover a larger fraction of the basin, producing greater discharge. Because the threshold represents the set of (D, I) pairs that generate the same hydrological response, even for D > Tc a decrease in Ic required to maintain that discharge is observed, resulting in a negatively sloped threshold. Accordingly, considering durations longer than Tc remains physically meaningful in this setting, because the evolving areal coverage of precipitation cells can increase discharge over time. Therefore, for a fixed target discharge (i.e., the same hydrological response), the critical intensity Ic required to achieve it decreases as duration increases.

A significant number of nighttime events (C0) could not be classified due to the limited performance of the infrared illumination system available at the UNIBO station. This limitation would not have occurred if the classification had relied on images from the Hortus stations, which are equipped with a more robust power supply capable of supporting higher-performance infrared lighting. This highlights that one of the main constraints of employing a lightweight and low-cost monitoring station is the reduced image quality during nighttime hours.

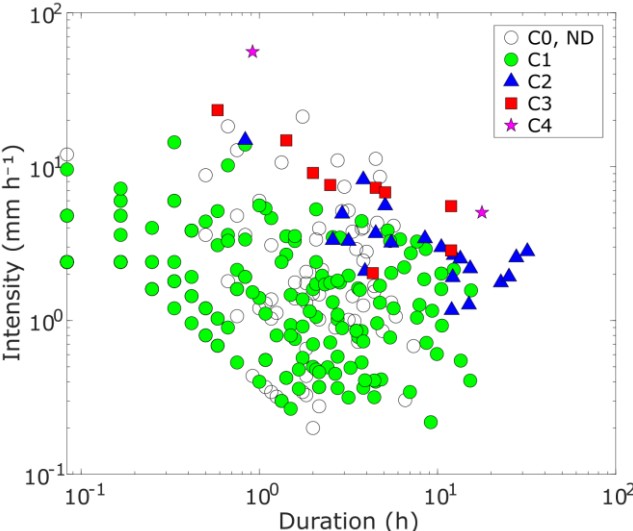

**Figure 9: Rainfall duration vs. mean intensity for precipitation recorded by the UNIBO gauge. Both axes are on a logarithmic scale.**

### 4.2 Rainfall thresholds

The two thresholds defined to separate the hydrological responses observed at the UNIBO station, namely TH1, which separates low-flow events (C1) from those causing an increase in water level (C2, C3, C4), and TH2, which isolates debris flows (C4) from all other classes (C1, C2, C3), are shown in Figure 10. The performance metrics for TH1 indicate good classification capability, with an accuracy of 0.91, AUC of 0.87, and F1-score of 0.73. The false positive rate is 7.1 %, while the false negative rate is 18.8 %. It follows a power-law relationship described by the equation $I = \alpha \cdot D^{\beta}$, with a scaling exponent $\beta$ of −0.56 and a coefficient $\alpha$ of 7.44 (Table 3).

TH2 correctly identified the two debris-flow-triggering rainfall events without false positives, resulting in all performance metrics equal to 1.00. However, two unclassified events, and therefore not included in the dataset used for threshold calculation, were located above this threshold: one occurred during a period when the camera had not yet been installed, and the other took place at night. The scaling exponent $\beta$ of this threshold is −0.56, while the coefficient $\alpha$ is 22.04 (Table 3). This threshold is consistent with the regional threshold proposed by Ceriani et al. (1994) (THCER) for the Lombardia Region (Fig. 10b).

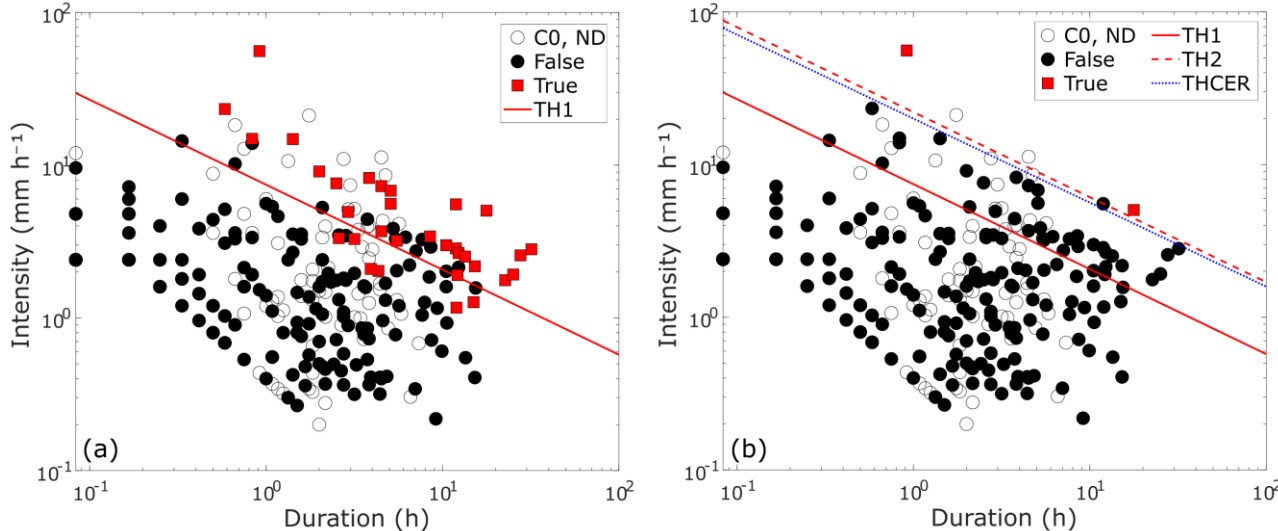

**Figure 10: Rainfall duration vs. mean intensity. (a) The "True" class includes events classified as high flow (C2), high flow with sediment transport (C3), and debris flows (C4), while the "False" class corresponds to low-flow events (C1). TH1 represents the LDA-derived threshold that separates C2, C3, and C4 from C1. (b) The "True" class includes only C4 events, with C1, C2 and C3 events classified as "False". TH2 is the shifted threshold identifying debris flows. THCER represents the regional threshold proposed by Ceriani et al. (1994) for the Lombardia Region. Both axes are on a logarithmic scale.**

The same methodology used for the UNIBO dataset was applied to precipitation data from the Hortus rain gauges (Fig. 11) to evaluate the sensitivity of the thresholds to the rain-gauge location. Across these sites, the scaling exponent β of the power law equation ranges from −0.44 to −0.64, while the coefficient α varies between 6.64 and 8.73. Accuracy consistently remains above 0.94, AUC above 0.88, and F1-score above 0.80, indicating robust overall performance (Table

3). TH1 calculated for H1 and H2, the gauges installed closest to the initiation area, successfully identified all C2, C3, and C4 events without false negatives, achieving the highest performance metrics within the network. Interestingly, while H1 displays threshold parameters (α and β) similar to those identified at UNIBO, H2 exhibits slightly higher intercept and scaling exponent values. The better performance of H1 and H2 may be attributed to the proximity of these gauges to the initiation area, likely allowing for more accurate capture of localized rainfall conditions. For TH2, all performance metrics

across the Hortus network are equal to 1.00.

The scaling exponent β, which has the same value for both TH1 and TH2, shows percentage variations at the Hortus stations with respect to UNIBO ranging from −14.29 % (H2) to +21.43 % (H4). The α coefficient varies relative to UNIBO from −10.75 % (H4) to +17.34 % (H3) for TH1, and from −30.85 % (H4) to +43.15 % (H2) for TH2. Considering a 30 min rainfall event, the critical rainfall for high-flow conditions (TH1) is 5.5 mm for UNIBO and H1, 6.4 mm for H2,

6.3 mm for H3, and 4.5 mm for H4. For debris flows (TH2), the 30 min critical rainfall is 16.2 mm for UNIBO, 17.7 mm for H1, 24.6 mm for H2, 15.9 mm for H3, and 10.3 mm for H4. These variations highlight the influence of gauge location on the estimated rainfall thresholds.

In addition, operator-based classification of events may have introduced uncertainty in the derived thresholds. To assess reproducibility, we ran a sensitivity analysis on 11 events whose classification was uncertain, 9 involving the transition

from low flow (C1) to high flow (C2) and 2 from high flow (C2) to high flow with sediment transport (C3). Each event was reassigned to its alternative plausible class, and the rainfall thresholds were recomputed. For the high-flow threshold, the refitted parameters are slope β = −0.51 and α = 8.34 (−7.9% and +12.0% relative to TH1). For the debris-flow threshold, the coefficient α is 19.74, corresponding to −10.5% relative to TH2 (β fixed equal to the high-flow threshold). For a 30-min storm, the associated critical rainfalls are 5.9 mm for high-flow conditions and 14.1 mm for debris-flow

initiation (compared to 5.5 mm for TH1 and 16.2 mm for TH2). These uncertainties are minor relative to the overall uncertainty sources, and our conclusions remain robust even under a worst-case reclassification in which all 11 ambiguous events were simultaneously reassigned to their alternative plausible class.

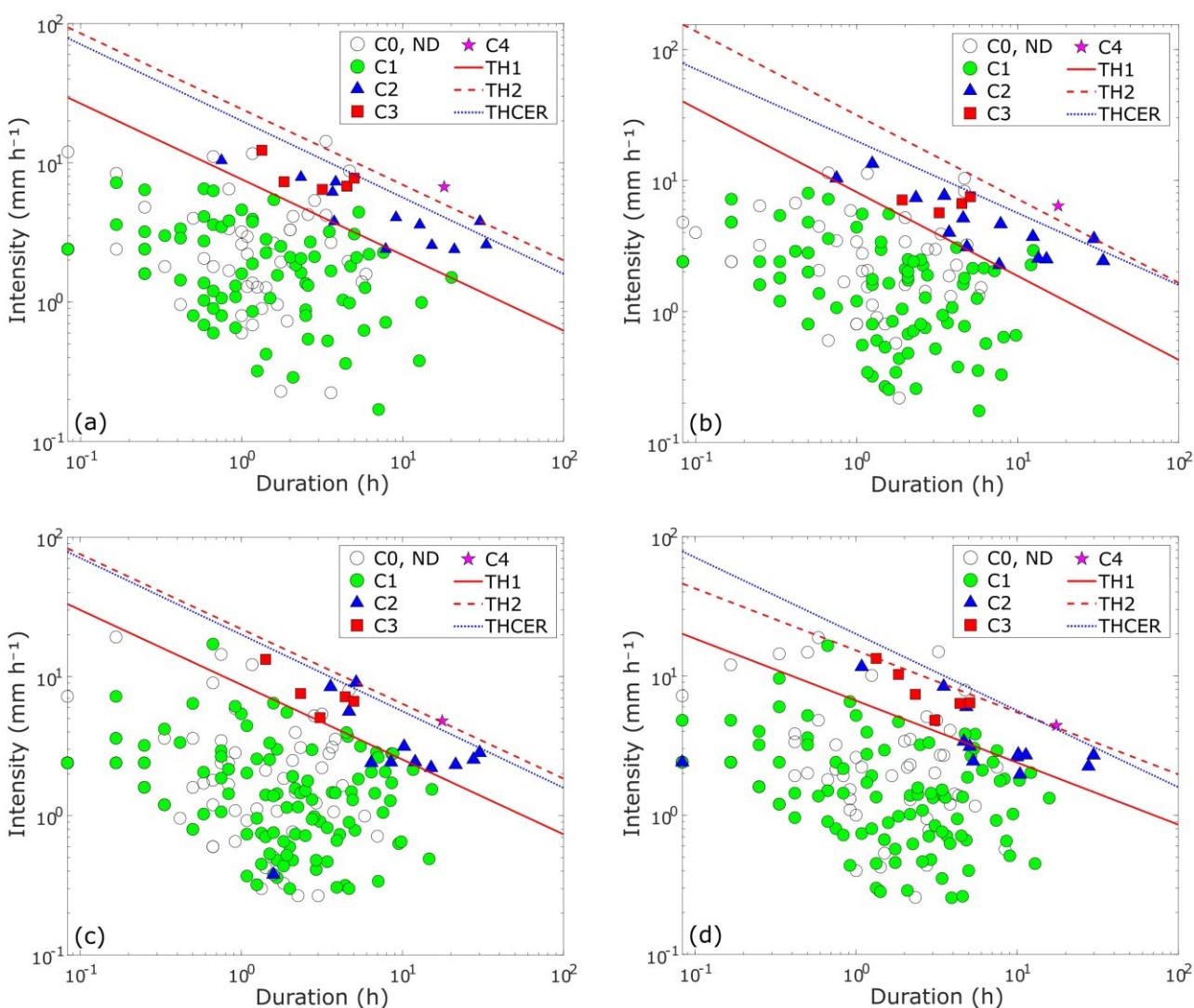

**Figure 11: Rainfall duration vs. mean intensity. Each chart corresponds to a different rain gauge: (a) H1, (b) H2, (c) H3, and (d) H4. Axes are on logarithmic scale.**

**Table 3. Summary of the parameters α and β (derived from the equation I = α · D $^β$) for TH1 and TH2, reported for each rain gauge. Model performance is evaluated using three metrics: accuracy (Acc), area under the ROC curve (AUC), and F1-score (F1). Performance metrics of TH2 are all equal to 1.000.**

| | TH1 | | | | | TH2 | |
|---|---|---|---|---|---|---|---|
| | α | β | Acc | AUC | F1 | α | β |
| **UNIBO** | 7.44 | -0.56 | 0.912 | 0.871 | 0.732 | 22.04 | -0.56 |
| **H1** | 7.61 | -0.54 | 0.970 | 0.960 | 0.919 | 24.31 | -0.54 |
| **H2** | 8.15 | -0.64 | 0.981 | 0.988 | 0.950 | 31.55 | -0.64 |
| **H3** | 8.73 | -0.54 | 0.955 | 0.902 | 0.811 | 21.93 | -0.54 |
| **H4** | 6.64 | -0.44 | 0.944 | 0.884 | 0.800 | 15.24 | -0.44 |

## 5. Discussion

### 5.1 Impact of spatial variability of rainfall on threshold estimation

The results of this study indicate that monitoring data can effectively support the definition of rainfall thresholds for debris-flow initiation. Including all rainfall events within a relatively short observation period, not just the critical ones,
enables a more robust identification of the rainfall conditions associated with different hydrological responses. However, the findings also highlight the importance of the monitoring station's location. In our case, the scaling exponent β and the α coefficient of the debris-flow threshold (TH2) vary by 18.5 % and 29.8 %, respectively, when the rain-gauge position is shifted within the catchment by less than 150 meters, the distance between H1 and H2 gauges. The largest percent changes in intensity among the five rain gauges for the duration of 30 minutes is 137.8 %, while for the duration of 1 hour
is 107.0 %, in both cases observed between gauges H2 and H4, with H2 reporting higher intensities than H4.

A direct comparison between the UNIBO station and the Hortus stations located upslope (H1, H2, H3) and downslope (H4) of UNIBO provides insight into this critical aspect. Using Deming errors-in-variables regression with equal error variances on both axes (Francq and Govaerts, 2014), we compared the Hortus measurements with those from the UNIBO reference. Figure 12 presents differences in precipitation amount (a), duration (b), and mean intensity (c) reporting for
each gauge the 95% confidence intervals estimated via a nonparametric bootstrap.

Precipitation totals show that the 95% confidence bands for H1 and H2 lie above the 1:1 line, whereas H3 largely overlaps the line and H4 lies below it. This pattern is consistent with an elevation effect: at higher elevations (H1, 1,330 m; H2, 1,248 m), greater precipitation depths are measured, whereas at lower elevations (H3, 770 m; H4, 695 m) totals are close to or smaller than those recorded at UNIBO (Fig. 12a).

For rainfall duration (Fig. 12b), the 95% confidence bands for H2 and H3 overlap with the 1:1 line, indicating good agreement with the reference gauge. H1 tends to overestimate event duration compared to UNIBO, while H4 tends to underestimate it. The band for H1 is also wider than the others, reflecting greater uncertainty in the relationship between H1 and the reference gauge. In contrast, H4 shows a narrower confidence band, indicating a tighter relationship with the reference gauge.

For rainfall intensity (Fig. 12c), the 95% confidence bands for H2 consistently lie below the 1:1 line, while those for H1 also fall below the line for UNIBO intensities exceeding 8 mm h$^{-1}$. This indicates that both H1 and H2 tend to record lower rainfall intensities compared to the reference gauge, despite generally measuring greater precipitation depths. This discrepancy is likely explained by longer event durations recorded at these stations, which reduce the computed mean intensity. H3 and H4 show intensity measurements consistent with those recorded at the UNIBO station, although the
confidence bands are comparatively wide, indicating appreciable inter-event variability rather than a systematic bias.

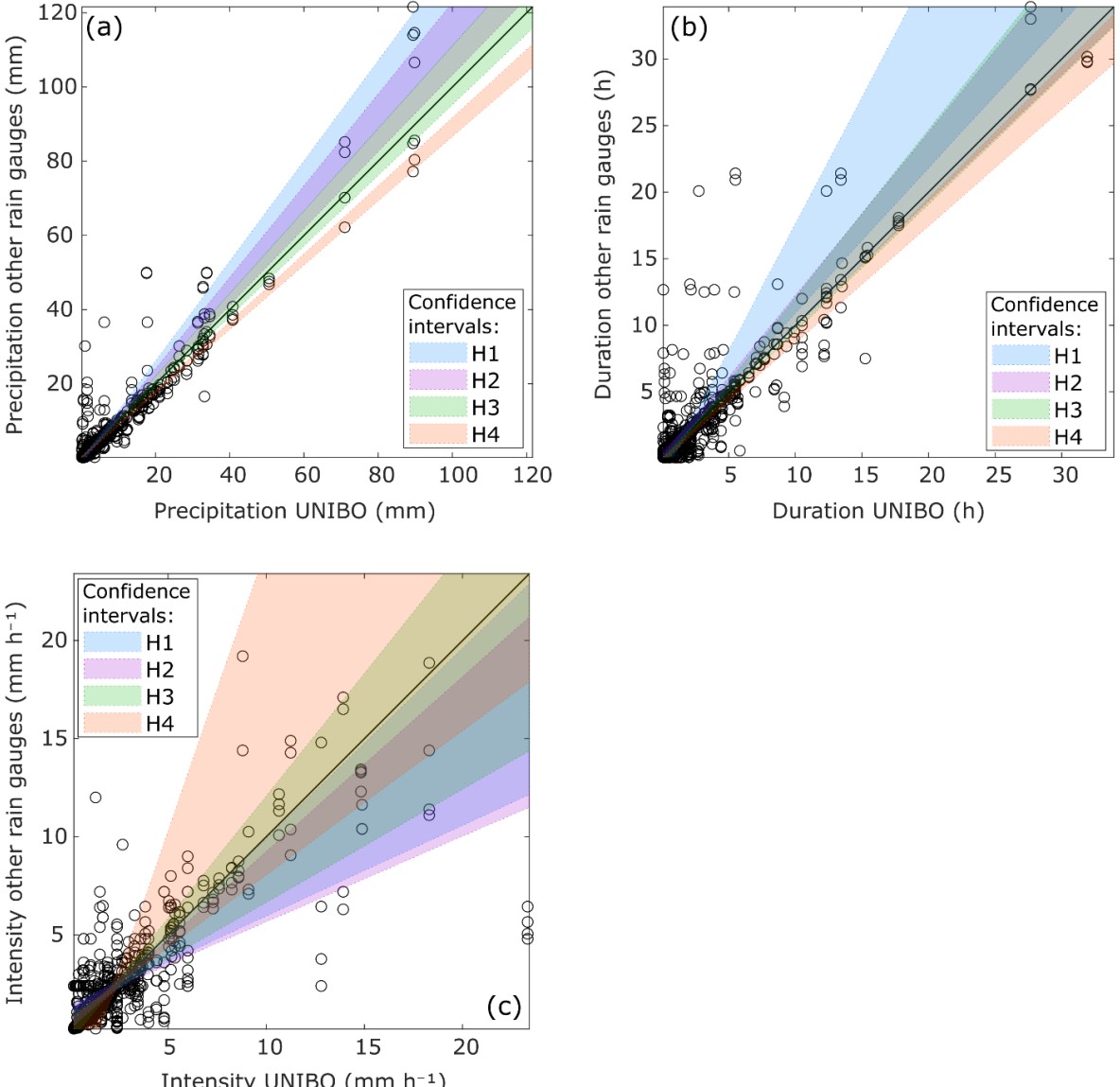

**Figure 12: Comparison of rainfall event characteristics recorded at the UNIBO rain gauge and each of the four Hortus stations, where each point represents a single event measured at both locations. (a) Precipitation; (b) duration; (c) mean intensity. Shaded bands show 95% confidence intervals for the fitted Deming relationships ($\lambda=1$), obtained via nonparametric bootstrap (2,000 resamples). The black 1:1 line indicates perfect equivalence.**

One way to visualize the impact of spatial rainfall variability on the definition of rainfall thresholds is through the intensity–duration plot. Figure 13 shows the classified rainfall events recorded at the UNIBO station and at least one additional Hortus station, along with dispersion bars representing the range of duration and intensity values measured by the rain gauges for each event (see Section 3.5, Fig. 7). As can be seen, some rainfall events exhibit narrow ranges, indicating consistent measurements among stations, whereas others reveal a broader spread, pointing to significant spatial variability in precipitation across the basin. The shorter dispersion bars observed for high-intensity rainfall events result from the log-log scaling, which compresses differences at higher values while stretching them at lower ones, resulting in a reduced visual representation of spatial variability at the upper end of the scale.

To further explore this aspect, 10,000 random simulations were performed via a bootstrap procedure with replacement: for each rainfall event recorded by UNIBO and at least one additional Hortus station, one of the gauges that recorded the event (UNIBO, H1, H2, H3, or H4) was randomly selected, and its paired (D, I) values were taken. For each simulation,

the corresponding TH1 threshold was calculated using LDA, resulting in an uncertainty band (Uncertainty Band for Rainfall variability, UBR; Fig. 13b) given by the envelope of all thresholds. The spread of the simulated thresholds reflects the variability introduced by spatial distribution of rainfall. Notably, this variability band consistently excludes the mean values of debris-flow events (C4) and high-flow events with sediment transport (C3), suggesting that the impact of rainfall spatial variability on threshold definition is moderate. However, it must be noted that the Blè catchment is small compared to the typical size of the storm cells that trigger debris flows in the region, and the two most distant stations (H1 and H4) are separated by less than 1,450 metres. Other studies have shown that significant differences in rainfall measurements can occur when rain gauges are placed farther from the source area (Abancó et al., 2016; Marra et al., 2014; Marra et al., 2016; Nikolopoulos et al., 2014). This highlights the importance of measuring rainfall within the catchment and, ideally, as close as possible to debris-flow initiation zones.

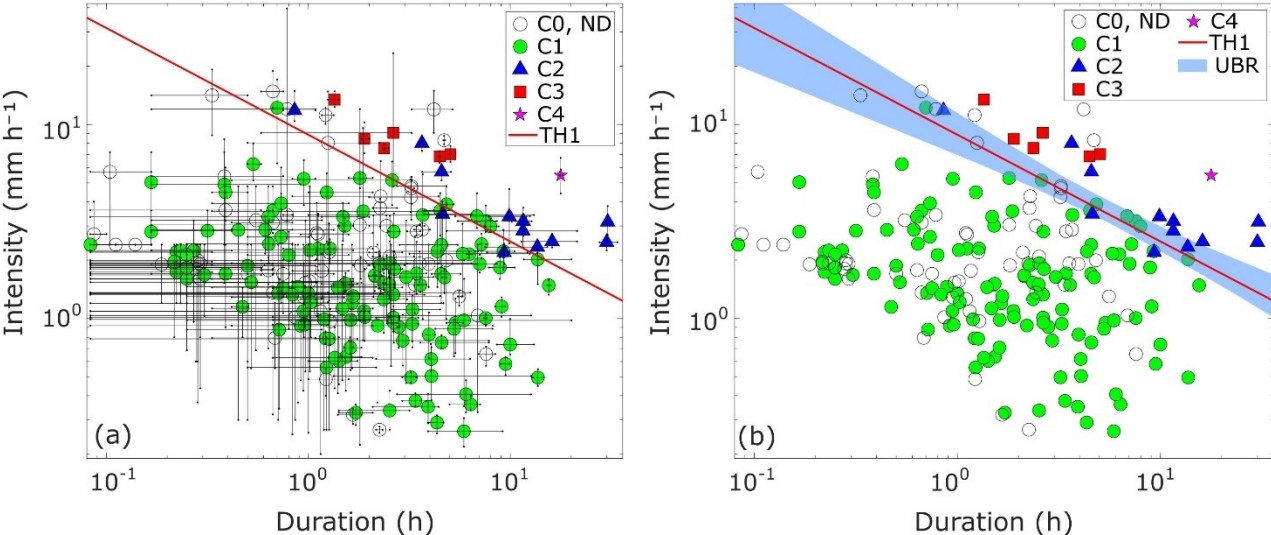

**Figure 13: (a) Ranges (min–max bars) and mean values (dots) of rainfall duration and intensity for each event recorded by UNIBO and at least one additional Hortus station. Values are calculated across five stations: UNIBO, H1, H2, H3, and H4. Note that, due to the logarithmic scale of both axes, bar lengths are also represented on a logarithmic scale. (b) The blue band (UBR) represents the envelope of TH1 curves derived from 10,000 bootstrap simulations, selecting a random gauge per event and taking its paired (D, I). The red line TH1, shown in both panels, is the threshold calculated using UNIBO data for rainfall events that were also recorded by at least one Hortus station (monitoring periods 2022 and 2023 only).**

## 5.2 Impact of the criterion used to define rainfall events

Another potential source of uncertainty in estimating rainfall thresholds lies in the criterion used to define rainfall events. As described in Section 3.2, we defined an event as a period of continuous rainfall with no more than 2 hours without precipitation (i.e., more than 0 mm recorded within each 2-hour window). Using a different criterion would directly influence the calculated duration and intensity of rainfall events, particularly during complex episodes composed of multiple bursts, and could result in different threshold estimates.

To assess the impact of this factor, we redefined the TH1 threshold by varying the minimum dry period used to separate rainfall events, testing intervals of 1, 3, 4, 5, 6, 7, 8, 12, 18, and 24 hours. These thresholds were used to construct an uncertainty band (Uncertainty Band for Event definition, UBE; Fig. 14).

The width of this uncertainty band is comparable to that derived from rainfall spatial variability (Fig. 13b). The rainfall thresholds exhibit varying scaling exponents $\beta$ and coefficients $\alpha$ of the power-law equation, with no systematic

relationship to dry-period duration (Table 4). No single threshold consistently outperforms the others across all evaluation metrics. The highest classification accuracy (0.914) was achieved with a 3-hour separation, while the area under the ROC curve (AUC) peaked at 0.879 with a 4-hour separation. The F1-score, which emphasizes the correct identification of positive cases, reached its maximum (0.800) with a 12-hour separation, as this interval isolates longer and more cumulative rainfall events. However, longer dry periods also reduce the average rainfall intensity of events by including low-intensity phases within the same episode. As a result, even actual triggering events may appear with lower intensity in the intensity–duration space, potentially overlapping with non-triggering events. Therefore, a dry-period duration of 3 to 4 hours represents a reasonable compromise for the Blè catchment, as it preserves the distinction of triggering events and minimize distortion of their rainfall characteristics.

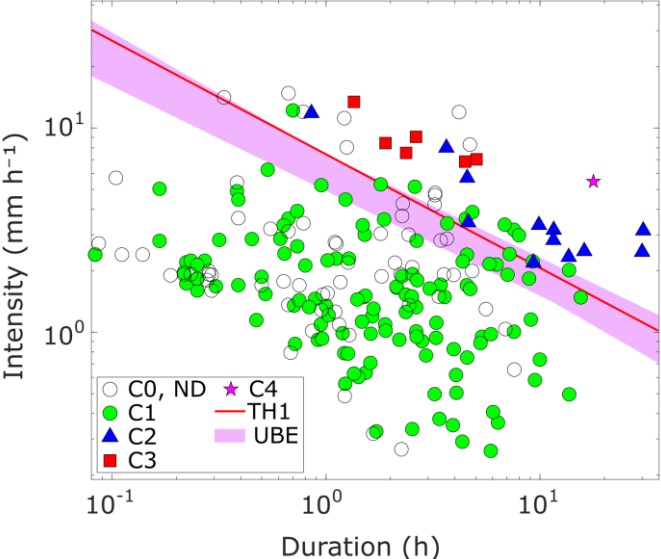

**Figure 14: Mean values (dots) of rainfall duration and intensity for each event recorded by UNIBO and at least one additional Hortus station. The pink band UBE (Uncertainty Band for Event definition) represents the range of LDA-derived thresholds calculated using dry-period durations between events of 1, 2, 3, 4, 5, 6, 7, 8, 12, 18, and 24 hours. TH1 is the threshold calculated with all the UNIBO dataset and a dry period of 2 hours.**

**Table 4: Summary of the parameters α and β (from the equation I = α · D^β) for TH1, calculated on UNIBO data using different dry-period durations between events. Model performance is evaluated using three metrics: accuracy (Acc), area under the ROC curve (AUC), and F1-score (F1).**

| D [h] | α | β | Accuracy | AUC | F1 |
|-------|------|-------|----------|-------|-------|
| 1 | 7.42 | -0.51 | 0.890 | 0.761 | 0.571 |
| 2 | 7.44 | -0.56 | 0.912 | 0.871 | 0.732 |
| 3 | 7.13 | -0.54 | 0.914 | 0.871 | 0.758 |
| 4 | 7.29 | -0.59 | 0.904 | 0.879 | 0.765 |
| 5 | 7.39 | -0.58 | 0.892 | 0.857 | 0.739 |
| 6 | 7.33 | -0.58 | 0.895 | 0.872 | 0.758 |
| 7 | 7.15 | -0.59 | 0.890 | 0.857 | 0.750 |
| 8 | 7.37 | -0.60 | 0.891 | 0.846 | 0.754 |
| 12 | 5.81 | -0.53 | 0.906 | 0.865 | 0.800 |
| 18 | 4.92 | -0.51 | 0.866 | 0.853 | 0.780 |
| 24 | 6.15 | -0.61 | 0.848 | 0.848 | 0.786 |

Overall, the criterion used to define rainfall events also appears to have a moderate influence on the position of the threshold in the intensity–duration plot. Naturally, combining uncertainties from both the event definition and the spatial variability of rainfall could lead to greater overall uncertainty. However, in our specific case, the critical rainfall events associated with debris flows and high sediment transport remain clearly identifiable, indicating that the analysis is robust.

### 5.3 Hydrological interpretation of rainfall thresholds

A major strength of our method, which relies on monitoring data from many rainfall events, is the ability to identify thresholds not only for debris-flow initiation but also for earlier stages of hydrological response. The lower threshold, TH1, which separates events that do not change channel flow depth from those that cause a measurable increase, is particularly relevant from a hydrological standpoint. It marks the point at which rainfall surpasses the catchment's initial losses, producing overland flow on exposed rock surfaces and shallow runoff along talus-slope drainage lines, and ultimately supplying water to the main debris-flow channel. This empirical threshold can be further supported by a simple hydrological analysis that improves understanding of catchment response.

Figure 15 compares the UNIBO TH1 threshold with the theoretical runoff discharge computed at the UNIBO monitoring station using the SCS Curve Number (CN) rainfall-excess model combined with the SCS dimensionless Unit Hydrograph (CN-UH method; Soil Conservation Service, 1972). A similar approach was applied by Gregoretti et al. (2016) and Berti et al. (2020) to evaluate rainfall excess in debris-flow initiation zones of alpine catchments. Input data for the analysis are listed in Table 5. The key parameter of the method is the Curve Number, which defines the watershed's potential maximum retention and directly controls runoff generation. We derived a composite CN as the area-weighted average of three values assigned to exposed bedrock, old landslides, and debris deposits that characterize the basin upstream of the UNIBO station. A sensitivity analysis was carried out using minimum and maximum CN values for each unit, derived from USDA-SCS lookup tables and from values back-calculated by Bernard et al. (2025) for three monitored basins in the Eastern Italian Alps. All analyses assumed normal antecedent moisture conditions (AMC II), and the time of concentration was estimated with Kirpich's formula (Kirpich, 1940). Berti et al. (2020) and Bernard et al. (2025) used an initial abstraction Ia = 0.1S (where S is the potential maximum retention), calibrated on the known hydrological response

of the catchment. In this study, the standard SCS-CN value of Ia = 0.2S was adopted due to the lack of this basin-specific hydrological information.

The results show a fairly good agreement between empirical and theoretical thresholds. In particular, this agreement is clear for high CN values, which reflect low infiltration capacity. In these cases, the zero-discharge line marking the onset of channel runoff coincides with the lower boundary of the blue triangles, which indicate visible increases in flow depth recorded on video. Nevertheless, the theoretical TH1 threshold is steeper than the empirical one. This discrepancy arises from the simplified assumptions of the SCS-CN abstraction model. As highlighted by Berti et al. (2020), under the assumption of constant initial loss the model behaves like a simple "bucket," where the catchment begins to spill once its storage capacity is filled. In such conditions, the theoretical slope of the runoff-initiation threshold is –1, compared with –0.56 for the empirical threshold. The gentler empirical slope suggests that initial losses increase with rainfall duration, likely due to long-term infiltration into weathered rock or debris, an effect not represented in this simplified analysis.

With regard to the empirical debris-flow threshold (TH2), the model indicates that debris mobilization corresponds to a peak runoff discharge of about 2–3 m³/s. These values appear much higher than the critical surface discharge values reported by Gregoretti and Dalla Fontana (2008) and Berti et al. (2020) in similar geological settings, which are typically below 0.2 m³/s. However, it should be emphasized that the runoff discharges in Fig. 15 are computed at the UNIBO station, not in the initiation area, where the contributing headwater catchment is considerably smaller. More relevant to our analysis is the fact that the empirical threshold TH2 is roughly parallel to a theoretical line of equal-runoff discharge, again supporting the physical basis of the threshold identified from monitoring data. Although the discharge contours do not exactly match the slope of the runoff-initiation line, the discrepancy is minor and difficult to detect in empirical datasets. Consequently, the simplified assumption of slope similarity between TH1 and TH2 remains theoretically founded.

**Table 5. Parameters adopted for the SCS-CN and SCS Unit Hydrograph (SCS–UH) analysis at the UNIBO monitoring station. The table reports basin descriptors, land-cover/soil units with corresponding Curve Numbers (CN), and hydrological parameters used for runoff and hydrograph computation.**

| Parameter | | Value |
|---|---|---|
| Basin characteristics | Basin area (m$^2$) | 2052904 |
| | Basin length (m) | 2856 |
| | Basin height (m) | 1800 |
| Land cover | Rock area (m$^2$) | 1245455 |
| | Landslide area (m$^2$) | 328718 |
| | Debris area (m$^2$) | 478731 |
| | Rock Curve Number [min–max] | 85-95 |
| | Landslide Curve Number [min–max] | 60-70 |
| | Debris Curve Number [min–max] | 70-80 |
| | Composite Curve Number [min-max]] | 77-87 |
| Hydrological parameters | Potential Maximum Retention, S (mm) [min-max] $S=(25400/CN)-254$ | 38-76 |
| | Initial Abstractions, Ia (mm) [min-max] $Ia=0.2S$ | 8-15 |
| | Time of Concentration, Tc (h) *from Kirpich's formula* | 0.18 |

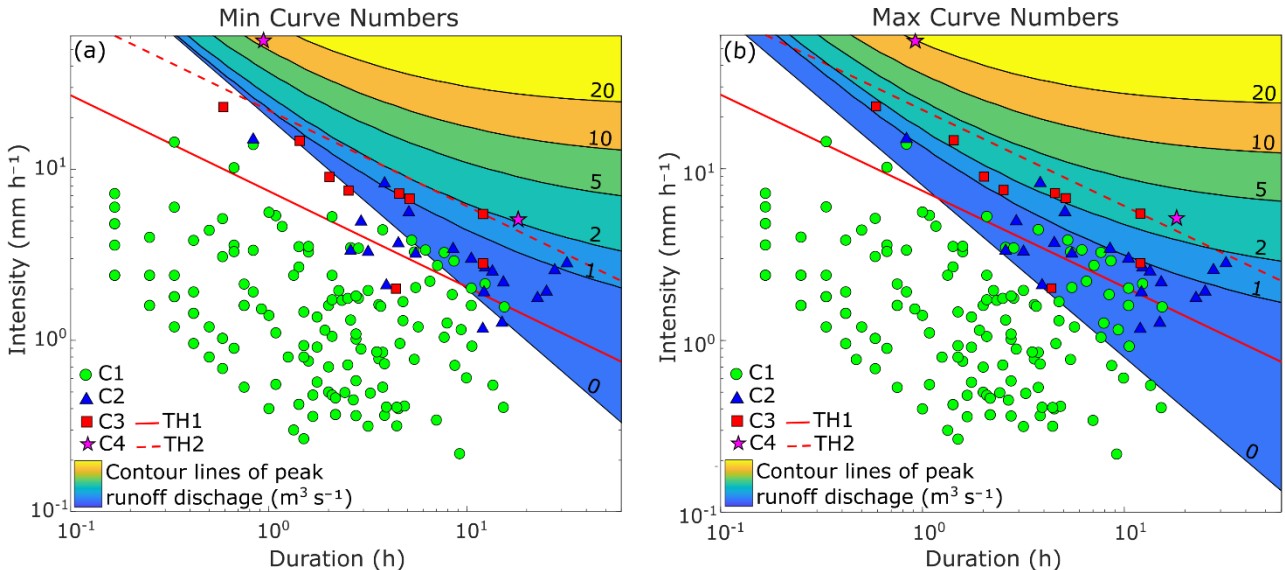

**Figure 15. Contour maps of peak runoff discharge obtained with the SCS–UH method for (a) minimum CN values and (b) maximum CN values. Empirical observations of rainfall events are superimposed, with symbols indicating event classification. The comparison illustrates the sensitivity of theoretical runoff estimates to Curve Number selection.**

## 6. Conclusions

Rainfall thresholds for the initiation of debris flows were defined using monitoring data collected in a catchment in the Alps over three summer seasons. The analysis focused not only on distinguishing debris-flow from non-debris-flow conditions but also on identifying hydrological thresholds associated with increased streamflow and sediment transport. This was achieved using a simple, low-cost, and easy-to-deploy monitoring station.

The main results of the study are as follows:

- Rainfall events associated with high flow (C2) and high flow with sediment transport (C3) generally had longer durations and higher intensities compared to non-responsive conditions (C1), although some overlap in the intensity–duration scatterplot was observed.

- The two rainfall events that triggered debris flows were clearly distinguishable from the rest due to their specific combinations of intensity and duration. One event was a typical short and intense summer storm, while the other was an autumnal rainfall.

- Between 30 % and 40 % of rainfall events occurred at night and could not be classified due to the limited visibility of the infrared illumination system supported by the power supply of our lightweight station.

- Thresholds TH1 and TH2 showed good performance in identifying hydrological responses. The dual-threshold approach enables discrimination between different levels of catchment response and offers potential as a pre-alert tool.

- The impact of spatial variability in rainfall across the catchment is moderate, as the variability band of the simulated TH1 thresholds consistently excludes the mean values of debris-flow (C4) and high-flow with sediment transport (C3) events. This is likely due to the relatively short distances between the rain gauges, which are often smaller than the characteristic spatial scale of convective storm cells responsible for triggering such events in the region.

• The choice of the dry period used to distinguish two consecutive events had a moderate influence. A dry period of 3 or 4 hours appears to offer a good compromise in this catchment, preserving event distinctiveness without distorting rainfall characteristics.

The approach proposed in this study can be effectively applied to catchments where debris flows are triggered by channel bed erosion and exhibit periodic, rainfall-driven activity. Its simplicity and low cost make it a practical tool for defining

rainfall thresholds and supporting early warning efforts in data-scarce environments.

**Data availability**

The data are freely available from the corresponding author upon request.

**Author contribution**

EI: Conceptualisation, Methodology, Formal analysis and investigation, Writing - original draft preparation. MR: Data

collection and material preparation, Writing - review and editing. ER: Data collection and material preparation, Writing - review and editing. AS: Data collection and material preparation, Writing - review and editing. LB: Data collection and material preparation, Writing - review and editing. MC: Data collection and material preparation, Writing - review and editing. MB: Conceptualisation, Data collection and material preparation, Methodology, Formal analysis and investigation, Writing - review and editing, Funding acquisition, Resources, Supervision. All authors read and approved

the final manuscript.

**Competing interests**

The authors declare that they have no conflict of interest.

**Financial support**

This work was funded by the Lombardia Region under the project "Dynamics of debris flows in Val Camonica valley

(Brescia): field monitoring of the Val Rabbia and Blè debris-flow catchments" and within the RETURN Extended Partnership, which received funding from the European Union Next-GenerationEU (National Recovery and Resilience Plan – NRRP, Mission 4, Component 2, Investment 1.3 – D.D. 1243 2/8/2022, PE0000005).

**Declaration of AI and AI-assisted technologies in the writing process**

During the preparation of this work the authors used ChatGPT 4 (chat.openai.com) to enhance the grammar and syntax,

as well as to refine the sentence structure. All the content is original, and no concepts, ideas, or interpretations were produced by this tool. After using this tool, the authors reviewed and edited the content as needed and take full responsibility for the content of the publication.

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
