# Peer review of "Identification of rainfall thresholds for debris-flow occurrence through field monitoring data."

_EGUsphere, 2025_

## Author Comment (AC1)

Dear Editor and Reviewer,

Thank you very much for your useful comments and suggestions.

In this document, you will find a detailed explanation of the changes made to the original manuscript to meet your suggestions.

For the sake of clarity, we used the following text styles:

| | |
|---|---|
| black, italics: | reviewer comment |
| blue, plain text: | our reply |
| *blue, italics:* | revised text |

Best regards

Elena Ioriatti

Mauro Reguzzoni

Edoardo Reguzzoni

Andreas Schimmel

Luca Beretta

Massimo Ceriani

Matteo Berti

1. The authors position the main innovation of this manuscript as an alternative method for defining rainfall thresholds that relies on monitoring data collected over a relatively short period and does not require extensive records of debris-flow events. This claim suggests the use of a physics-based approach, which typically does not demand large event datasets. However, the thresholds in this study appear to be derived empirically, which generally does require a substantial number of events for development and validation. Could the authors clarify why they believe their method circumvents the need for numerous debris-flow events? Further elaboration on this point would be helpful.

We regret the lack of clarity in our previous description and thank you for pointing this out. The thresholds were derived empirically with the aim of collecting as much empirical data as possible within a limited monitoring period. Since intense events are rare in such a short timeframe, we adopted two strategies:

1. We included all the recorded rainfall events, not only the triggering but also the non-triggering ones. To this end, we applied Linear Discriminant Analysis, which defines the threshold through the statistical separation of predefined classes.

2. We considered as triggering not only debris-flow events (C4) but also all events that produced a basin response, i.e. high flow (C2) and high flow with sediment transport (C3) and we calculated the lower threshold TH1. As the hydrological process is directly linked to discharge, different I–D combinations along the same threshold correspond to the same flow type. This allows us to rigidly shift TH1 upward, keeping the slope constant and adjusting the intercept, until identifying TH2, which only isolates the debris-flow events.

These clarifications have been included in the Introduction of the revised manuscript as follows:

*In this study, we propose an alternative approach to define Intensity-Duration rainfall thresholds, which is based on the use of monitoring data collected over a relatively short period and does not require a large number of debris-flow events. The method relies on data acquired through relatively low-cost sensors and a lightweight, easy-to-install monitoring station. This station was located on the stream bank within an Alpine catchment. The monitoring data provided a good understanding of the catchment's hydrological response, allowing the identification of a lower threshold associated with increases in stream water level and sediment transport that may serve for pre-alert purposes.*

*To overcome the limitation posed by the small number of debris-flow events, we propose two complementary strategies. First, we consider not only triggering but also non-triggering rainfall events, applying statistical analysis to distinguish between the two classes. Second, we draw on the larger set of high-flow and sediment-transport events to establish a robust lower threshold, which then serves as a reference for isolating debris-flow conditions and defining the debris-flow threshold.*

*In addition, the study explores the uncertainty in threshold definition associated with two key factors: the spatial location of the rain gauge and the duration of the inter-event time used to separate rainfall events.*

2. Another highlighted contribution is the incorporation of the catchment hydrological regime, with the classification of four regimes: C0, C1, C2, C3, and C4. However, the classification criteria seem somewhat arbitrary and based on expert judgment. To strengthen the robustness of this classification, it is recommended to incorporate quantitative hydrologic variables—such as water level or runoff—given that hydrological regimes are fundamentally governed by these factors. Since water level data are already monitored at stations H1 and H2, integrating these measurements into regime classification is advisable. Alternatively, employing a hydrological model to simulate runoff across different regimes could help elucidate the underlying mechanisms.

Thank you for this suggestion. We examined the water-level sensor data, but unfortunately they proved unreliable. For some events an increase is visible in the radar dataset, while for others it is not, and the dataset also contains negative values. The sensor is suspended high above the channel to avoid damage during debris

flows, and its wide measurement cone is appropriate for capturing extreme events but not sensitive enough to reliably detect smaller flow increases. The attached image illustrates this issue: during a high-flow event (C2) recorded at the Hortus station, a small rivulet is visible below the radar but it flows along one bank of the channel rather than in the center, making accurate measurement difficult. For this reason, we could not integrate water-level measurements into the regime classification.

[Figure]

**3.3 Classification of images and linking rainfall events to the observed channel response**

*[…]*

*Data from the water level sensor were examined but proved unreliable, as the dataset contained negative values and showed inconsistencies with the observed flow dynamics. The sensor was mounted high above the channel to avoid damage during debris flows, and its wide measurement cone is suitable for capturing extreme events but insufficiently sensitive to reliably detect smaller flow increases.*

We thank you for the suggestion of using a hydrological model: in the revised manuscript we applied the SCS-CN and Unit Hydrograph approach to simulate runoff and compare the results with the empirical thresholds. This new analysis and discussion have been added in Sect. 5.3 and are described in detail in our response to the following comment.

3. The manuscript dedicates significant space to discussing the effects of spatial differences and minimum inter-event time on rainfall thresholds. While relevant, these aspects have been explored in previous studies. It is suggested to condense this section and instead expand the discussion on the influence of hydrological regimes on rainfall thresholds, which represents a more novel aspect of this work.

Thank you for this valuable suggestion. A new analysis has been performed to interpretate the thresholds in relation to hydrological regimes (new Sect. 5.3). This section now links the empirical thresholds to runoff generation modelled using the SCS-CN and Unit Hydrograph approach and discusses the physical basis for TH1 and TH2.

**3.4 Rainfall threshold definition using the Linear Discriminant Analysis (LDA)**

*[…]*

*For the debris-flow threshold (TH2), events classified as debris flows (C4) were treated as "triggering" ("True"), while all other classes (C1, C2, C3) were treated as "non-triggering" ("False"). The LDA method*

*was not applied in this case because the limited number of debris flows made it difficult to reliably estimate within-class variance and class means for a stable discriminant axis. Moreover, the strong imbalance between classes biases 
[revised manuscript text omitted]

4. In Fig. 8, should the legend indicate the hydrological regimes C0, C1, C2, C3, and C4? Please verify and revise as necessary.

Thank you for noticing the error in the legend. The figure has been corrected.

[Figure]

5. For Fig. 3, please add a legend identifying the monitoring stations H1, H2, H3, etc.

Thank you for the suggestion. The icons for the stations have been added to the legend.

[Figure]

New references:

Bernard, M., Barbini, M., Berti, M., Boreggio, M., Simoni, A., and Gregoretti, C.: Rainfall-Runoff Modeling in Rocky Headwater Catchments for the Prediction of Debris Flow Occurrence, Water Resources Research, 61, e2023WR036887, https://doi.org/10.1029/2023WR036887, 2025.

Berti, M., and Simoni, A.: Experimental evidences and numerical modelling of debris flow initiated by channel runoff, Landslides, 2, 171-182, https://doi.org/10.1007/s10346-005-0062-4, 2005.

Berti, M., Bernard, M., Gregoretti, C., and Simoni, A.: Physical Interpretation of Rainfall Thresholds for Runoff-Generated Debris Flows, J. Geophys. Res. Earth Surf., 125, https://doi.org/10.1029/2019JF005513, 2020.

Gregoretti, C. and Fontana, G. D.: The triggering of debris flow due to channel-bed failure in some alpine headwater basins of the Dolomites: analyses of critical runoff, Hydrol. Process., 22, 2248–2263, https://doi.org/10.1002/hyp.6821, 2008.

Gregoretti, C., Degetto, M., Bernard, M., Crucil, G., Pimazzoni, A., De Vido, G., Berti, M., Simoni, A., and Lanzoni, S.: Runoff of small rocky headwater catchments: Field observations and hydrological modeling, Water Resour. Res., 52, 8138–8158, https://doi.org/10.1002/2016WR018675, 2016.

Kirpich, Z. P.: Time of concentration of small agricultural watersheds, Civ. Eng., 10, 362, 1940.

Simoni, A., Bernard, M., Berti, M., Boreggio, M., Lanzoni, S., Stancanelli, L. M., and Gregoretti, C.: Runoff-generated debris flows: Observation of initiation conditions and erosion–deposition dynamics along the channel at Cancia (eastern Italian Alps), Earth Surf. Process. Landforms, 45, 3556–3571, https://doi.org/10.1002/esp.4981, 2020.

Soil Conservation Service (SCS): Section 4: Hydrology, in: National Engineering Handbook, U.S Department of Agriculture, Washington DC, 1972.

---

## Author Comment (AC2)

Dear Editor and Dr. Francesco Marra,

Thank you very much for your useful comments and suggestions.

In this document, you will find a detailed explanation of the changes made to the original manuscript to meet your suggestions.

For the sake of clarity, we used the following text styles:

| | |
|---|---|
| black, italics: | reviewer comment |
| blue, plain text: | our reply |
| blue, italics: | revised text |

Best regards

Elena Ioriatti

Mauro Reguzzoni

Edoardo Reguzzoni

Andreas Schimmel

Luca Beretta

Massimo Ceriani

Matteo Berti

Line 35: I think the reference to Nikolopoulos & al here is misplaced as this paper does not aim at developing or testing empirical thresholds.

Thank you for your observation. The reference to Nikolopoulos et al. 2014 is not fully aligned with the purpose of this sentence. In the revised manuscript, we have removed this citation.

Lines 43-49: Temporal resolution of the rainfall data also constitutes an important factor for empirical thresholds (Marra & al 2019; Gariano & al, 2020).

Thank you for your valuable suggestion. We have included temporal resolution as an additional factor influencing the reliability of rainfall thresholds in the revised text.

*The reliability of rainfall thresholds defined with an empirical approach can be influenced by several sources of uncertainty, including the spatial distribution of rain gauges, the criteria used to define individual rainfall events, and the temporal resolution of rainfall data. Marra et al. (2016) and Nikolopoulos et al. (2014) highlighted that rain gauge networks with limited spatial coverage can underestimate rainfall during convective storms. This may lead to thresholds that do not accurately reflect triggering conditions. Another key source of uncertainty lies in the method used to identify discrete rainfall events from continuous data (Melillo et al., 2015). A common approach is to use a minimum inter-event time, but there are still no clear criteria for determining its optimal duration (Dunkerley, 2008). The temporal resolution of rainfall data is another important factor, as coarse resolution has been shown to systematically underestimate depth–duration (ED) thresholds (Marra, 2019; Gariano et al., 2020).*

Furthermore, we have specified the temporal resolution of our rainfall data at lines 150–151 (Sect. 3.3):

*Rainfall data with a temporal resolution of 5 minutes were collected between late spring and early autumn in 2021, 2022, and 2023.*

Lines 149-150: This is an unnecessary level of detail for a basic analysis. It sounds more like a technical report than a scientific paper.

Thank you for your comment. The sentence has been removed in the revised manuscript.

Lines 173-176: it seems that the classification is still done by an operator. I suggest removing this and simply state that the classification was done based on an operator.

Thank you for the comment. We have clarified that event classification was performed by an operator. We also consider it important to specify that a script with a user interface was employed to automatically record the start and end times of each observed process in a structured dataset, as this detail is relevant for replicating the method. Indeed, performing the classification manually and transcribing the start and end times of each process is time-consuming and prone to errors.

*Event classification was operator-based and supported by a script that displays the images and automatically generates a structured dataset with class labels and corresponding time intervals. In an initial attempt, classification was carried out manually by reviewing each image, recording start and end times, and assigning class labels. This method proved inefficient and error-prone due to manual transcription. The script improved the process by providing an interface with simple controls that allowed the operator to classify each image, automatically creating a continuous time-series dataset that recorded the class and the corresponding start and end times of each observed process. Although classification still required expert input, the script greatly reduced transcription errors and accelerated the overall workflow.*

Lines 182-183: "the operator became more adept…" does this mean the quality of the classification changes over time? What are the implications for the analyses? Would two different operators do the same classification, would we get the same results? How would these potential differences affect the AUC? Would a sensitivity analysis to these subjective choices help quantifying the uncertainties related to the proposed method?

Thank you for raising this point and for noting the potential ambiguity of that sentence. We recognise that image classification inevitably involves a degree of subjectivity. To minimise this, we defined classes to be as objective as possible, while acknowledging that the analysis necessarily remains operator based. To reduce this subjectivity, two of the authors jointly examined a large number of cases and established shared classification criteria. Following this initial training, these criteria were applied consistently across all images, thereby limiting operator bias. We did not perform a sensitivity analysis, as the classification was considered sufficiently robust.

*Some classification uncertainties arose during rainfall, when visibility was compromised by dense fog or water droplets on the camera lens. In such instances, it was sometimes helpful to examine images of the channel before and after the event to determine whether sediment transport, along with associated erosion or deposition, had occurred. Weather conditions, such as sunny or cloudy days and the time of day, could also lead to misinterpretations. For example, in shaded areas water flow was often less visible, while in bright sunlight reflections on the water surface could create the illusion of higher discharge. To address these challenges, two of the authors jointly reviewed a large number of cases and established shared criteria to distinguish actual changes in discharge from lighting or visibility artefacts. These criteria were then applied consistently across all images to ensure comparability and limit operator bias.*

Lines 191-192: again, this is an unnecessary level of detail for a basic analysis

Thank you for the suggestion. The sentence has been modified removing that we used a custom script.

*Each precipitation event was characterised by the highest class observed during its duration.*

Section 3.4: it is not mentioned what features are examined for this dimensionality reduction. This is crucial information. It turns out from section 4.2 that basically this is a 2-dimensional clustering with duration and average intensity. Should be stated here.

Thank you for this valuable comment. We have revised the text to explicitly state that rainfall duration and average intensity were defined a priori as the predictor variables.

*In this study, we did not apply dimensionality reduction from a larger feature set; instead, rainfall duration and average intensity were defined a priori as the predictor variables for the analysis.*

Lines 219-223: if I understand this correctly, this means that the threshold was optimised to maximise the AUC. I don't understand why the 0.1 steps in log scale are needed for this. Seems like a lot of details for an optimisation.

We apologise for the oversight and thank you for your comment. We have revised the text to clarify that TH2 is a new threshold, specifically defined to identify debris flows, and not an optimisation of TH1. TH2 was obtained by shifting the threshold TH1 upward along the intensity axis with increments of $\alpha$ of 0.1, while maintaining the same slope $\beta$. The threshold TH2 was selected as the one, among all candidates generated at each 0.1 step, that maximised AUC. The choice of 0.1 increments was a practical way to explore the threshold space with sufficient resolution.

*Two thresholds were defined to separate the observed hydrological responses: TH1 is a lower threshold distinguishing low flow (C1) from high flow, high flow with sediment transport and debris flow (C2, C3, C4), while TH2 is an upper threshold distinguishing debris flow (C4) from all other classes (C1, C2, C3).*

*[…]*

*TH2 was derived by keeping the scaling exponent β (slope) of TH1 and iteratively increasing the coefficient α in increments of 0.1, which in the log–log form of $I = α \cdot D^β$ corresponds to shifting the intercept $\log_{10} α$ upward. The model performance was evaluated at each step by using the Area Under the Receiver Operating Characteristic Curve (AUC). The final threshold was selected as the value that maximized AUC, ensuring the best separation between debris flows and non-debris flows.*

Figure 7: this is trivial, the text in 3.5 is enough to understand that you computed the min, max and mean values across the different monitoring stations.

We agree that the information in Section 3.5 already describes the procedure; however, we would prefer to keep Figure 7 as it makes it immediately clear that for each rainfall event recorded by the UNIBO rain gauge and at least one Hortus rain gauge, we calculated the minimum, maximum, and mean values. These values were then used to plot the error bars shown in Figure 13.

Section 4.2 lots of methods here. The first sentence (lines 277-279) is a repetition of what stated in section 3.4 and should be removed. Lines 280-285 instead provide important information on the methods that should be moved to section 3.4.

Thank you for the suggestion. We have removed these sentences from the Results section and revised Section 3.4 to provide this information more effectively.

*3.4 Rainfall threshold definition using the Linear Discriminant Analysis (LDA)*

*Two thresholds were defined to separate the observed hydrological responses: TH1 is a lower threshold distinguishing low flow (C1) from high flow, high flow with sediment transport and debris flow (C2, C3, C4), while TH2 is an upper threshold distinguishing debris flow (C4) from all other classes (C1, C2, C3). To determine TH1, the method of Linear Discriminant Analysis (LDA) was applied to the dataset, treating low-flow events (C1) as non-triggering ("False") and all other responses as triggering (C2, C3, C4, "True"). LDA is a statistical method for dimensionality reduction and feature selection that identifies a linear combination of input variables to optimally separate triggering and non-triggering classes (Fisher, 1936; Ramos-Cañón et al., 2016). In this study, we did not apply dimensionality reduction from a larger feature set; instead, rainfall duration and average intensity were defined a priori as the predictor variables for the analysis. LDA was then applied to the rainfall events, with the aim of identifying a discriminant axis that maximizes between-class variance and minimizes within-class variance, as described by the objective function J(w) in Eq. (1) […]*

It would be very useful here to know that is the advantage of using this LDA clustering over other methods for defining thresholds that are more common in literature. In particular, the way TH2 is calculated resembles a lot the frequentist approach by Brunetti et al 2010 in which a slope in log(D)-log(I) coordinates is kept constant and the intercept is changed to match some condition (here to optimize the AUC and there to leave a pre-defined proportion of observed events below). In both cases, the question on whether the same slope should be used is a fundamental one. Perhaps it should be discussed in the frame of this method: what is the hydrological reasoning behind using the same slope?

The main reason we employed LDA rather than the frequentist approach of Brunetti et al. (2010) is the limited number of events available, which did not allow for a robust probabilistic analysis of the type used in their

study. LDA offers an alternative by defining the threshold through statistical separation of predefined classes, which can be applied even with relatively small datasets.

As for the question of using the same slope for the two thresholds, previous studies on runoff-generated debris flows (e.g. Berti and Simoni, 2005; Simoni et al., 2020; Berti et al., 2020) have shown that thresholds tend to display similar slopes, at least for the short-duration events typical of debris-flow initiation. This reflects the fact that both runoff generation and debris mobilization are controlled by the same hydraulic process, namely the concentration of overland flow and its transformation into channelized flow. This point has been clarified in Section 3.4 of the Methods. Moreover, new analyses have been performed on the hydrological interpretation of the two thresholds using the SCS Curve Number (CN) rainfall–excess model combined with the SCS dimensionless Unit Hydrograph (CN–UH method; Soil Conservation Service, 1972) and the results added as Section 5.3 of the Discussion.

**3.4 Rainfall threshold definition using the Linear Discriminant Analysis (LDA)**

*[…]*

*For the debris-flow threshold (TH2), events classified as debris flows (C4) were treated as "triggering" ("True"), while all other classes (C1, C2, C3) were treated as "non-triggering" ("False"). The LDA method was not applied in this case because the limited number of debris flows made it difficult to reliably estimate within-class variance and class means for a stable discriminant axis. Moreover, the strong imbalance between classes biases 
[revised manuscript text omitted]

Figures 9, 10 and 11: it is notable that the separation between C1 events and other events is better at short durations and then the different events merge for longer durations. One could claim this is because longer-duration events likely include short-duration bursts with higher intensities that determine the hydrological response. What is the time of concentration of the catchment? Given the fact that 2 hours separation are considered enough for separating events (and, therefore, antecedent conditions are relatively negligible), does it make sense to average intensities over durations longer than the time of concentration? It would be useful commenting on this aspect.

Many thanks for the insightful comment. The time of concentration (Tc) range between 11 min (Kirpich) and 17 min (Giandotti). For all rainfall events that showed a basin response (C2, C3, C4), we extracted the burst and recalculated the intensity. It should be noted that the extraction of the burst introduces an additional degree of subjectivity. The figure shows the new data in magenta, red, and blue, while the corresponding previously identified events are shown in transparency. We observed that discrimination among classes does not improve; in particular, classes C2 and C3 remain mixed, as was the case with the previously identified events in Fig. 9 of the manuscript. Although in theory, for durations longer than Tc, keeping the intensity constant, discharge should be constant (the triggering intensity should remain constant, i.e., the threshold would be horizontal for D>Tc), in practice this is unlikely because rainfall is not uniformly distributed across the basin. Therefore, as duration increases, it is more likely that the rainfall cell moves and covers a larger portion of the basin, producing a greater discharge. Because the threshold represents the set of (I, D) pairs that generate the same hydrological response, even for D > Tc a decrease in the critical intensity required to maintain that discharge is observed, resulting in a negatively sloped threshold. Accordingly, considering durations longer than Tc remains physically meaningful in this setting, because the evolving areal coverage of precipitation cells can increase discharge over time. Therefore, for a fixed target discharge (i.e., the same hydrological response), the critical intensity Ic required to achieve it decreases as duration increases.

[Figure]

*4.1 Classified rainfall events*

*[...]*

*The separation of C2, C3, and C4 events from C1 events is clearer at short durations than at longer durations. One could argue that this occurs because long-duration events include short high-intensity phases (bursts) that control the hydrological response, and that it is therefore not appropriate to consider durations exceeding the time of concentration (Tc). For durations close to Tc the entire catchment contributes to runoff; consequently, at the intensity Ic associated with Tc, durations greater than Tc should not further increase discharge. One would therefore expect the threshold to be horizontal for D > Tc, indicating an approximately constant basin response.*

*In practice, however, rainfall is not uniformly distributed across the catchment. As duration increases, the precipitation cell is more likely to shift and cover a larger fraction of the basin, producing greater discharge. Because the threshold represents the set of (I, D) pairs that generate the same hydrological response, even for D > Tc a decrease in Ic required to maintain that discharge is observed, resulting in a negatively sloped threshold. Accordingly, considering durations longer than Tc remains physically meaningful in this setting, because the evolving areal coverage of precipitation cells can increase discharge over time. Therefore, for a fixed target discharge (i.e., the same hydrological response), the critical intensity Ic required to achieve it decreases as duration increases.*

Lines 334-336: in addition to the percent change of beta e alfa, it would be useful to know the largest percent changes in the intensities for the range of durations that are considered useful for the triggering in the area.

Following the proposed approach, we computed the largest percentage changes in rainfall intensity among the five gauges for durations considered relevant for landslide triggering in the study area. Specifically, for D = 0.5 h the maximum percentage change is 137.8%, while for D = 1 h it is 107.0%. These values have now been reported in the revised manuscript.

*The largest percent changes in intensity among the five rain gauges for the duration of 30 minutes is 137.8%, while for the duration of 1 hour is 107.0%.*

Figure 12: how are the regressions and the related uncertainties computed? Usual linear regression models assume no error on the variable used in the x axis and homoschedastic variables on the y axis, which is not necessarily the case here, since there is complete symmetry between UNIBO and the other stations.

We decided to adopt the UNIBO rain gauge as the reference, considering it as the ground truth and assessing all comparisons against it. We now state this choice explicitly in the revised manuscript.

*Rainfall data recorded at the Hortus stations were compared against the UNIBO reference (treated as ground truth) using ordinary least-squares linear regression for each rainfall-event characteristic. Figure 12 illustrates the differences in precipitation amount (a), duration (b), and mean intensity (c) between the reference UNIBO station and the four Hortus stations, which are positioned upslope (H1, H2, H3) and downslope (H4). Each plot includes the 95% confidence bands of the regression lines, shown in different colours for each rain gauge.*

Line 350: how is this statistical significance calculated?

Thank you for pointing this out. The earlier statement incorrectly inferred statistical significance from whether the plotted confidence band appeared to exclude the 1:1 line. We have corrected this mistake in the text.

*For the mean intensity, when UNIBO intensity exceeds 15 mm h⁻¹, the 95% confidence band for H1–H3 lies entirely below the 1:1 line, suggesting a systematic negative bias. By contrast, the band for H4 still overlaps the 1:1 line, indicating approximate agreement also at higher intensities (Fig. 12c).*

We also corrected this part of the manuscript:

*In general, total precipitation measurements show good agreement between UNIBO and Hortus rain gauges. For higher cumulative totals, the 95% confidence bands for H1 and H2 lie predominantly above the 1:1 line, whereas H3 overlaps to it and H4 lies below it. These patterns are consistent with an elevation effect: at higher altitudes (H1, 1,330 m; H2, 1,248 m) the fitted mean lies above the 1:1 line, whereas at lower altitudes (H3, 770 m; H4, 695 m) it is close to or below the 1:1 line (Fig. 12a).*

*The 95% confidence bands for H1, H2, and H3 overlap the 1:1 line across the observed range, suggesting approximate agreement in event duration relative to the UNIBO gauge, whereas the band for H4 does not overlap the 1:1 line for event durations longer than 25 hours. The band for H1 is wider than the others, indicating greater uncertainty in how event durations at H1 relate to UNIBO, whereas H4 shows a narrower band, suggesting a tighter relationship (Fig. 12b).*

Lines 368-370: how are these random samples taken? Uniform distributions over x and y? Normal distributions? Is the correlation between I and D considered? Since I is calculated from D, there is a correlation between the variables that must be accounted for in such an analysis (lower D in one station implies that higher I is more likely than lower I, etc.). ID and ED thresholds are equivalent from several points of view, but not from this one. I believe the blue area UBR in Figure 13 cannot be interpreted, and the conclusion that "the impact of spatial variability on the threshold definition is moderate" cannot be stated unless the points above are clarified and, if necessary, amended.

Random samples are taken following a uniform distribution over *x* and *y* because we consider all precipitation values within the maximum–minimum duration and maximum–minimum intensity intervals across the five rain gauges to be equally probable. Although there is indeed a correlation between duration and intensity, we believe that the blue area (UBR) is still valid because the random samples are drawn from within a distribution of real measured data. This important information on the distribution and the correlation between the variables has been added to the revised manuscript.

*To further explore this aspect, 10,000 random simulations were performed, in which for each rainfall event a random point was sampled within the uncertainty rectangle defined by the minimum and maximum observed values of duration and intensity. The samples were generated following a uniform distribution, so that each value within these ranges had the same probability of being selected. For each simulation, the corresponding TH1 threshold was calculated using the LDA method, resulting in an uncertainty band (Uncertainty Band for*

*Rainfall variability, UBR; Fig. 13b). Although duration and intensity are correlated, this approach ensures that the sampling still reflects the variability captured in the measured data, since all random points are confined to the ranges derived from the observations.*

New references:

Bernard, M., Barbini, M., Berti, M., Boreggio, M., Simoni, A., and Gregoretti, C.: Rainfall-Runoff Modeling in Rocky Headwater Catchments for the Prediction of Debris Flow Occurrence, Water Resources Research, 61, e2023WR036887, https://doi.org/10.1029/2023WR036887, 2025.

Berti, M., and Simoni, A.: Experimental evidences and numerical modelling of debris flow initiated by channel runoff, Landslides, 2, 171-182, https://doi.org/10.1007/s10346-005-0062-4, 2005.

Berti, M., Bernard, M., Gregoretti, C., and Simoni, A.: Physical Interpretation of Rainfall Thresholds for Runoff-Generated Debris Flows, J. Geophys. Res. Earth Surf., 125, https://doi.org/10.1029/2019JF005513, 2020.

Gariano, S. L., Melillo, M., Peruccacci, S., and Brunetti, M. T.: How much does the rainfall temporal resolution affect rainfall thresholds for landslide triggering?, Nat Hazards, 100, 655–670, https://doi.org/10.1007/s11069-019-03830-x, 2020.

Gregoretti, C. and Fontana, G. D.: The triggering of debris flow due to channel-bed failure in some alpine headwater basins of the Dolomites: analyses of critical runoff, Hydrol. Process., 22, 2248–2263, https://doi.org/10.1002/hyp.6821, 2008.

Gregoretti, C., Degetto, M., Bernard, M., Crucil, G., Pimazzoni, A., De Vido, G., Berti, M., Simoni, A., and Lanzoni, S.: Runoff of small rocky headwater catchments: Field observations and hydrological modeling, Water Resour. Res., 52, 8138–8158, https://doi.org/10.1002/2016WR018675, 2016.

Kirpich, Z. P.: Time of concentration of small agricultural watersheds, Civ. Eng., 10, 362, 1940.

Marra, F.: Rainfall thresholds for landslide occurrence: systematic underestimation using coarse temporal resolution data, Nat Hazards, 95, 883–890, https://doi.org/10.1007/s11069-018-3508-4, 2019.

Simoni, A., Bernard, M., Berti, M., Boreggio, M., Lanzoni, S., Stancanelli, L. M., and Gregoretti, C.: Runoff-generated debris flows: Observation of initiation conditions and erosion–deposition dynamics along the channel at Cancia (eastern Italian Alps), Earth Surf. Process. Landforms, 45, 3556–3571, https://doi.org/10.1002/esp.4981, 2020.

Soil Conservation Service (SCS): Section 4: Hydrology, in: National Engineering Handbook, U.S Department of Agriculture, Washington DC, 1972.

---

## Author Response (AR2)

Dear Editor and Reviewers,

Thank you very much for your useful comments and suggestions.

In this document, you will find a detailed explanation of the changes made to the original manuscript to meet your suggestions.

For the sake of clarity, we used the following text styles:

| | |
|---|---|
| black, italics: | reviewer comment |
| blue, plain text: | our reply |
| blue, italics: | revised text |

Best regards

Elena Ioriatti

Mauro Reguzzoni

Edoardo Reguzzoni

Andreas Schimmel

Luca Beretta

Massimo Ceriani

Matteo Berti

1. Line 67: the term "propose" here can be misleading since the use of non-triggering events is already established in literature. I suggest "use" or similar terms.

Thank you for your comment, we have changed *'propose'* to *'use'*. However, we wish to note that this approach is common for historical data at larger scales but remains uncommon in catchment-scale monitoring.

*To overcome the limitation posed by the small number of debris-flow events, we use two complementary strategies. First, we consider not only triggering but also non-triggering rainfall events, applying statistical analysis to distinguish between the two classes. While this practice is established at regional scales, it has rarely been implemented within catchment-scale monitoring. Second, we draw on the larger set of high-flow and sediment-transport events to establish a robust lower threshold, which then serves as a reference for isolating debris-flow conditions and defining the debris-flow threshold.*

2. Lines 202-205: I still think the reproducibility of the study can be affected by this approach. This should be declared in the manuscript, and the potential impacts on the results should be discussed or evaluated.

Thank you for your the comment. To assess reproducibility we performed a sensitivity analysis on 11 events with uncertain classification, nine involving the transition from low flow (C1) to high flow (C2), and two from high flow (C2) to high flow with sediment transport (C3). We reassigned each to the alternative plausible class and recomputed the rainfall thresholds; the results under this worst-case reassignment are reported in the revised manuscript. We provide below the figure showing this alternative classification and comparisons between these thresholds and TH1/TH2.

[Figure]

*4.2 Rainfall thresholds*

*[…]*

*In addition, operator-based classification of events may have introduced uncertainty in the derived thresholds. To assess reproducibility, we ran a sensitivity analysis on 11 events whose classification was uncertain, 9 involving the transition from low flow (C1) to high flow (C2) and 2 from high flow (C2) to high flow with sediment transport (C3). Each event was reassigned to its alternative plausible class, and the rainfall thresholds were recomputed. For the high-flow threshold, the refitted parameters are slope $\beta = -0.51$ and $\alpha = 8.34$ ($-7.9\%$ and $+12.0\%$ relative to TH1). For the debris-flow threshold, the coefficient $\alpha$ is 19.74, corresponding to $-10.5\%$ relative to TH2 ($\beta$ fixed equal to the high-flow threshold). For a 30-min storm, the associated critical rainfalls are 5.9 mm for high-flow conditions and 14.1 mm for debris-flow initiation (compared to 5.5 mm for TH1 and 16.2 mm for TH2). These uncertainties are minor relative to the overall uncertainty sources, and our conclusions remain robust even under a worst-case reclassification in which all 11 ambiguous events were simultaneously reassigned to their alternative plausible class.*

3. Lines 252-256: this part is indeed an optimization (specifically, a maximization). I think the text here is still overly complex. One could write: "TH2 was set to have the same scaling exponent \beta (slope) of TH1, and the coefficient \alpha that maximises the Area Under the Receiver Operating Characteristic Curve (AUC). Although…"

Thank you for the comment, the lines have been revised in the manuscript.

*[…]*

*TH2 was defined by keeping the same scaling exponent $\beta$ (slope) as TH1 and selecting the coefficient $\alpha$ that maximises the Area Under the Receiver Operating Characteristic Curve (AUC). Although…*

4. Figure 12: I agree with the authors when they say that the UNIBO gauge was treated as ground truth. But what I question is whether this is appropriate here, given that rainfall spatial variability plays a major role (and not only the measurement accuracy of the gauge). I think a regression method that allows for uncertainty in both the variables should be considered. There are several available. Given the situation (spatial variability is important, likely the main factor), at a first approximation one could assume the same error variance in the two variables.

Thank you for this helpful suggestion. In the revision we re-fit all comparisons using Deming regression, which allows error in both axes. Consistent with your point that the gauges have comparable accuracy, we set the error-variance ratio to $\lambda = 1$. We quantified uncertainty via a nonparametric pairs bootstrap (2,000 resamples): BCa (Bias-Corrected and Accelerated) 95% confidence intervals were computed for the Deming slope and

intercept, while percentile 95% intervals were used to form the confidence bands of the regression line. Figure 12 and the corresponding text have been updated accordingly.

*5.1 Impact of spatial variability of rainfall on threshold estimation*

*[...]*

*A direct comparison between the UNIBO station and the Hortus stations located upslope (H1, H2, H3) and downslope (H4) of UNIBO provides insight into this critical aspect. Using Deming errors-in-variables regression with equal error variances on both axes (Francq and Govaerts, 2014), we compared the Hortus measurements with those from the UNIBO reference. Figure 12 presents differences in precipitation amount (a), duration (b), and mean intensity (c) reporting for each gauge the 95% confidence intervals estimated via a nonparametric bootstrap.*

*Precipitation totals show that the 95% confidence bands for H1 and H2 lie above the 1:1 line, whereas H3 largely overlaps the line and H4 lies below it. This pattern is consistent with an elevation effect: at higher elevations (H1, 1,330 m; H2, 1,248 m), greater precipitation depths are measured, whereas at lower elevations (H3, 770 m; H4, 695 m) totals are close to or smaller than those recorded at UNIBO (Fig. 12a).*

*For rainfall duration (Fig. 12b), the 95% confidence bands for H2 and H3 overlap with the 1:1 line, indicating good agreement with the reference gauge. H1 tends to overestimate event duration compared to UNIBO, while H4 tends to underestimate it. The band for H1 is also wider than the others, reflecting greater uncertainty in the relationship between H1 and the reference gauge. In contrast, H4 shows a narrower confidence band, indicating a tighter relationship with the reference gauge.*

*For rainfall intensity (Fig. 12c), the 95% confidence bands for H2 consistently lie below the 1:1 line, while those for H1 also fall below the line for UNIBO intensities exceeding 8 mm h$^{-1}$. This indicates that both H1 and H2 tend to record lower rainfall intensities compared to the reference gauge, despite generally measuring greater precipitation depths. This discrepancy is likely explained by longer event durations recorded at these stations, which reduce the computed mean intensity. H3 and H4 show intensity measurements consistent with those recorded at the UNIBO station, although the confidence bands are comparatively wide, indicating appreciable inter-event variability rather than a systematic bias.*

[Figure]

**Figure 1: Comparison of rainfall event characteristics recorded at the UNIBO rain gauge and each of the four Hortus stations, where each point represents a single event measured at both locations. (a) Precipitation; (b) duration; (c) mean intensity. Shaded bands show 95% confidence intervals for the fitted Deming relationships ($\lambda$=1), obtained via nonparametric bootstrap (2,000 resamples). The black 1:1 line indicates perfect equivalence.**

5. Line 441-449: please specify that the min-max value are min-max values among the ones observed by the available rain gauges (as done in the caption). Also, please specify how the UBR is obtained from these samples: is it the envelope of all the obtained values or is it some percentile interval? (Given the use of uniform distributions, I think percentile intervals should not be used here.) I suggest adding some caveats on this due to the independence assumption taken on I and D. One would expect to under-estimate variability when the two variables are assumed independent.

Thank you for the comment. To avoid this assumption of independence between D and I, we replaced the old method with a bootstrap-by-gauge scheme: for each event observed by UNIBO and at least one Hortus station,

we randomly select with replacement one of the gauges that recorded the event (UNIBO, H1–H4) and take that gauge's paired (*D,I*) values. This preserves the event-level dependence between duration and intensity and propagates inter-gauge variability. We recomputed TH1 for 10,000 bootstrap replicates and define the UBR as the outer envelope of all simulated thresholds (not a percentile band). Figure 13b and the manuscript text have been updated accordingly.

*5.1 Impact of spatial variability of rainfall on threshold estimation*

*[…]*

*To further explore this aspect, 10,000 random simulations were performed via a bootstrap procedure with replacement: for each rainfall event recorded by UNIBO and at least one additional Hortus station, one of the gauges that recorded the event (UNIBO, H1, H2, H3, or H4) was randomly selected, and its paired (D, I) values were taken. For each simulation, the corresponding TH1 threshold was calculated using LDA, resulting in an uncertainty band (Uncertainty Band for Rainfall variability, UBR; Fig. 13b) given by the envelope of all thresholds.*

[Figure]

Figure 2: (a) Ranges (min-max bars) and mean values (dots) of rainfall duration and intensity for each event recorded by UNIBO and at least one additional Hortus station. Values are calculated across five stations: UNIBO, H1, H2, H3, and H4. Note that, due to the logarithmic scale of both axes, bar lengths are also represented on a logarithmic scale. (b) The blue band (UBR) represents the envelope of TH1 curves derived from 10,000 bootstrap simulations, selecting a random gauge per event and taking its paired (D, I). The red line TH1, shown in both panels, is the threshold calculated using UNIBO data for rainfall events that were also recorded by at least one Hortus station (monitoring periods 2022 and 2023 only).

6. Figure 15: the blue triangles are hard to see over the blue background, I suggest changing the color of the symbols.

Thank you for the suggestion, the colours of the figure have been changed.

[Figure]

**Figure 15. Contour maps of peak runoff discharge obtained with the SCS–UH method for (a) minimum CN values and (b) maximum CN values. Empirical observations of rainfall events are superimposed, with symbols indicating event classification. The comparison illustrates the sensitivity of theoretical runoff estimates to Curve Number selection.**

**New References:**

Francq, B. G. and Govaerts, B. B.: Measurement methods comparison with errors-in-variables regressions. From horizontal to vertical OLS regression, review and new perspectives, Chemometrics and Intelligent Laboratory Systems, 134, 123–139, https://doi.org/10.1016/j.chemolab.2014.03.006, 2014.

The authors state: "To overcome the limitation posed by the small number of debris-flow events, we propose two complementary strategies. First, we consider not only triggering but also non-triggering rainfall events, applying statistical analysis to distinguish between the two classes."

However, it is standard practice—and indeed necessary—to derive rainfall thresholds using datasets that include both triggering and non-triggering rainfall events. Prior research has clearly shown that thresholds derived solely from triggering events tend to be unrealistically high. Therefore, I recommend removing the aforementioned statement to avoid overemphasizing what is already a widely accepted methodological requirement.

*Thank you for your comment; we have changed the verb 'propose' to 'use' so as not to overemphasise the novelty of this approach. However, we wish to highlight that in monitored basins it is uncommon to include non-triggering rainfall events, whereas this is common when non-triggering events are drawn from larger-scale historical series.*

*To overcome the limitation posed by the small number of debris-flow events, we use two complementary strategies. First, we consider not only triggering but also non-triggering rainfall events, applying statistical analysis to distinguish between the two classes. While this practice is established at regional scales, it has rarely been implemented within catchment-scale monitoring. Second, we draw on the larger set of high-flow and sediment-transport events to establish a robust lower threshold, which then serves as a reference for isolating debris-flow conditions and defining the debris-flow threshold.*

In the hydrological simulations (Table 5), the initial abstraction is set as Ia = 0.2S. Yet, several related studies conducted in alpine catchments—including Berti et al. (2020) and Bernard et al. (2025), both cited by the authors—adopt Ia = 0.1S. The discrepancy between these values and the authors' choice should be explicitly discussed, along with a justification for the selected parameter.

*Thank you very much for this insightful comment. In the studies conducted by Berti et al. (2020) and Bernard et al. (2025), the value Ia = 0.1S was calibrated using monitored events in which the hydrological response was known, thanks to the configuration of their monitoring sites, which included a sharp-crested weir measuring discharge at the outlet of the headwater catchment. In this study, since such information was not available, the commonly used value of Ia = 0.2S from the SCS Curve Number (CN) rainfall-excess model was adopted.*

*5.3 Hydrological interpretation of rainfall thresholds*

*[…] Berti et al. (2020) and Bernard et al. (2025) used an initial abstraction Ia = 0.1S (where S is the potential maximum retention), calibrated on the known hydrological response of the catchment. In this study, the standard SCS-CN value of Ia = 0.2S was adopted due to the lack of this basin-specific hydrological information.*